# Toward Unifying Group Fairness Evaluation from a Sparsity Perspective

## Abstract

Ensuring algorithmic fairness remains a significant challenge in machine learning, particularly as models are increasingly applied across diverse domains. While numerous fairness criteria exist, they often lack generalizability across different machine learning problems. This paper decouples existing fairness metrics and presents a unified framework through a sparsity perspective for measuring group fairness. This unified formulation can adapt to various machine learning problems such as classification and regression tasks. We demonstrate the effectiveness of the proposed sparsity-based framework as an evaluation metric through theoretical analysis and extensive experiments on a variety of datasets and bias mitigation methods. This work provides a novel perspective to algorithmic fairness by framing it through the lens of sparsity and social equity, offering potential for broader impact on fairness research and applications.

## 1 Introduction

Algorithmic fairness has been a key research challenge in machine learning. On the one hand, the growing adoption of automated machine learning models has streamlined decision-making in fields such as healthcare and finance. On the other hand, these models may produce biased and unfavorable outcomes for certain groups or individuals. This may be due to intrinsic biases in real-world datasets, which are influenced by societal biases against historically marginalized groups (Lee, 2018; Buolamwini & Gebru, 2018).

Research on mitigating model biases has led to the development of numerous fairness criteria and algorithms, along with extensive comparative studies evaluating their performance (Bellamy et al., 2018; Friedler et al., 2019; Wei et al., 2021; Alghamdi et al., 2022). However, most existing works focus on classification problems with binary targets (Agarwal et al., 2018; Zeng et al., 2022; Gaucher et al., 2023) or binary group membership (Denis et al., 2024), while fewer studies address fairness in regression problems (Agarwal et al., 2019; Chzhen et al., 2020; Xian et al., 2024). Moreover, the majority of existing group fairness notions are confined to a limited set of widely adopted criteria (Calders et al., 2009; Hardt et al., 2016; Agarwal et al., 2019), restricting their applicability to many real-world scenarios. Recent works by Alghamdi et al. (2022) and Xian et al. (2023) extend bias mitigation algorithms to multi-class problems and broaden the definition of existing fairness criteria. However, a universal fairness framework applicable across a broad range of machine learning problems remains absent.

Inspired by the Gini Index (Gini, 1912), widely used to evaluate sociological inequality in economics, and recent advances in sparsity measures (Diao et al., 2023), we connect algorithmic fairness assessment to the quantification of group vector sparsity by considering the full value distribution. In this work, we bridge the gap by introducing a unified, sparsity-based framework for quantifying algorithmic fairness. Our contributions can be summarized as follows:

1. For sparsity measures, we focus on the recently proposed PQ Index (Diao et al., 2023) and provide a theoretical guarantee of its effectiveness in measuring fairness (Section 3). Furthermore, we highlight its connection to the Maximum Pairwise Difference (MPD), a widely used method in algorithmic fairness, and the Gini Index, a well-established measure of inequality (unfairness) in economics.

2. We present a unified framework for measuring algorithmic fairness through decoupling of MPD and internal metrics from fairness evaluation and introduce novel measurements based on the sparsity of the full distribution of group-wise outputs (Section 4). This framework integrates existing fairness criteria and provides the flexibility for multi-group, multi-class, and regression problems, which are typically treated separately in existing studies.

3. To evaluate the performance of the proposed framework, we conduct extensive experiments across multiple datasets and bias mitigation techniques (Section 5). We demonstrate the framework's effectiveness by aligning the proposed metrics with established ones across benchmarks, and its broader applicability through analysis in intersectional fairness settings.

## 2 RELATED WORK

**Sparsity Measures.** Sparsity embodies the idea that a vector's magnitude is primarily determined by a few large components, reflecting inequality in the distribution of the vector components (Gini, 1912). It is important and widely utilized in various fields such as statistics and signal processing (Tibshirani, 1996; Donoho, 2006; Akçakaya & Tarokh, 2008). Various measures of sparsity have been proposed from different perspectives (Hurley & Rickard, 2009). The Gini Index (Gini, 1912) is a well-established measure of inequality in wealth or welfare distribution in economics (Dalton, 1920; Porath & Gilboa, 1994; Rickard & Fallon, 2004). Another type of sparsity measure is based on $\ell_p$ norms. For instance, $\ell_1$-norm-based constraints are frequently applied in function approximation (Barron, 1993), model regularization, and variable selection (Tibshirani, 1996; Chen et al., 2001). The PQ Index, defined as a ratio of $\ell_p$ norms, has been used for pruning deep neural networks (Hurley & Rickard, 2009; Diao et al., 2023). Motivated by the desirable properties of the PQ Index, this paper explores its theoretical foundations and applies it in the proposed fairness framework.

**Algorithmic Fairness.** Group fairness evaluates model predictions in relation to sensitive attributes. Among various group fairness criteria, statistical parity (Calders et al., 2009) and equalized odds (Hardt et al., 2016) are the most widely recognized in the fairness literature. In addition, statistical learning methods, including mutual information (Mary et al., 2019; Steinberg et al., 2020; Roh et al., 2020) and correlation (Beutel et al., 2019; Baharlouei et al., 2019; Grari et al., 2021), have been employed to quantify the extent of fairness violations. Han et al. (2023) recently proposed a distribution-level metric for classification using the total variation distance between the predicted probabilities of two sensitive groups. In contrast, our framework extends and unifies existing group-level fairness notions and can be applied to various machine learning problems.

**Bias Mitigation.** Fairness-promoting algorithms are generally categorized into three families (Angwin et al., 2016): pre-processing, in-processing, and post-processing. Pre-processing algorithms focus on transforming the data through feature editing (Feldman et al., 2015; Calmon et al., 2017) or reweighting (Kamiran & Calders, 2012). In-processing approaches consider a fair risk minimization problem and impose the fairness constraint during model optimization (Agarwal et al., 2018; Zhang et al., 2018; Baharlouei et al., 2019; Cho et al., 2020a; Lowy et al., 2021). Post-processing methods take in a biased base model and project its outputs to satisfy fairness constraints (Hardt et al., 2016; Kamiran et al., 2012; Pleiss et al., 2017). We include recent works under in-processing and post-processing to validate our proposed framework.

## 3 SPARSITY

Let $\boldsymbol{w} = [w_1, \ldots, w_d]^{\mathrm{T}} \in \mathbb{R}_+^d$ be a vector from the $d$-dimensional space of non-negative real numbers, where "T" denotes the transpose of a vector. Denote the values of the largest and smallest components of $\boldsymbol{w}$ by $w_{max}$ and $w_{min}$, respectively. Let $\mathbf{1}_d \triangleq [1, \ldots, 1]^{\mathrm{T}}$. A sparsity measure $S(\boldsymbol{w})$ quantifies the mass distribution among components of $\boldsymbol{w}$, with a larger value indicating higher sparsity. Existing fairness metrics often focus on measuring outcome gaps for the worst-case. A key observation is that many of these metrics can be decomposed into two components: a per-group evaluation metric and a Maximum Pairwise Distance (MPD) used for group comparisons. In this section, we explore the theoretical connections among the MPD and two sparsity measures, the Gini Index and the PQ Index.

**Definition 3.1** (Maximum Pairwise Difference). The Maximum Pairwise Difference of $\boldsymbol{w}$ is

$$MPD(\boldsymbol{w}) \triangleq \max_{i,j \in \{1,\ldots,d\}} |w_i - w_j|.$$

**Definition 3.2** (Gini Index). The Gini Index of $\boldsymbol{w}$ is

$$Gini(\boldsymbol{w}) \triangleq \frac{\sum_{i=1}^d \sum_{j=1}^d |w_i - w_j|}{2d \sum_{i=1}^d w_i}.$$

**Definition 3.3** (PQ Index). For any $0 < p < q$, the PQ Index of $\boldsymbol{w}$ is

$$\mathbf{I}_{p,q}(\boldsymbol{w}) = 1 - d^{\frac{1}{q} - \frac{1}{p}} \frac{\|\boldsymbol{w}\|_p}{\|\boldsymbol{w}\|_q},$$

where $\|\boldsymbol{w}\|_p = \left(\sum_{i=1}^d |w_i|^p\right)^{1/p}$ is the $\ell_p$-norm of $\boldsymbol{w}$ for any $p > 0$.

By definition, all the above sparsity measures attain their minimum value of 0 when the components of $\boldsymbol{w}$ are equal. Both the PQ Index and the Gini Index are scale-invariant in the sense that multiplying $\boldsymbol{w}$ by a positive factor does not change the values of $\mathbf{I}_{p,q}(\boldsymbol{w})$ or $Gini(\boldsymbol{w})$. However, the Maximum Pairwise Difference lacks this property.

**Relation to Fairness.** Fairness metrics in machine learning (Dwork et al., 2012; Hardt et al., 2016) and the Gini Index both reflect distributive justice (Everett & Everett, 2015), particularly Rawls' principle (Tao et al., 2014; Rawls, 2017). Originally an economic measure of inequality (Gini, 1912; Sen, 1997; Cowell, 2011), the Gini Index is well-suited to fairness analysis in machine learning due to its ability to capture disparities across the full distribution (Do & Usunier, 2022; Li et al., 2023a). However, criteria such as Statistical Parity (MPD among sensitive groups) (Alghamdi et al., 2022; Xian & Zhao, 2024) may overlook small-scale relative differences. The Gini Index and the PQ Index, which satisfy all six ideal sparsity properties (Hurley & Rickard, 2009), address this limitation by evaluating the entire output distribution, where higher sparsity indicates lower fairness.

### 3.1 PROPERTIES OF PQ INDEX

Diao et al. (2023) have shown that $0 \leq \mathbf{I}_{p,q}(\boldsymbol{w}) \leq 1 - d^{\frac{1}{q} - \frac{1}{p}}$, and a larger $\mathbf{I}_{p,q}(\boldsymbol{w})$ indicates a sparser vector. Additionally, PQ Index satisfies the six properties of an ideal sparsity measure proposed by Dalton (1920); Rickard & Fallon (2004). These properties require the sparsity to remain unchanged when $\boldsymbol{w}$ is multiplied by a positive scalar or appended by a duplicate. Moreover, adding a positive constant to each component of $\boldsymbol{w}$, or transferring $\alpha \in (0, (w_i - w_j)/2)$ from $w_i$ to $w_j$ where $i, j \in \{1, \ldots, d\}$ and $w_i > w_j$ reduce the sparsity. Additionally, appending $\boldsymbol{w}$ with a zero or adding a positive constant to a component that is sufficiently large will increase the sparsity. The details of them can be found in Appendix B.1.

In this work, we deepen the understanding of the PQ Index by providing the following theoretical properties. Theorems 3.1–3.4 characterize the properties of the PQ Index in quantifying fairness. Theorem 3.5 and 3.6 demonstrate connections and differences among MPD, Gini Index, and PQ Index. The proofs are presented in the Appendix B.2. The theorems bridges the proposed sparsity measure and fairness metrics commonly used in the machine learning literature, such as Statistical Parity (SP) and Equalized Odds (EO), both of which involve Maximum Pairwise Difference (MPD) as a component.

**Theorem 3.1.** *For $\boldsymbol{w}$, if there exists $k \in \{1, \ldots, d\}$ such that $w_k \neq 0$ and $w_j = 0$ for all $j \neq k$, then:*

$$\mathbf{I}_{p,q}(\boldsymbol{w}) = 1 - d^{\frac{1}{q} - \frac{1}{p}}$$

*Remark* 3.1. The above theorem indicates that the PQ Index reaches its maximum value when the vector contains only a single nonzero component.

**Theorem 3.2.** $\mathbf{I}_{p,q}(\boldsymbol{w})$ *is minimized if and only if $\boldsymbol{w} = c \cdot \mathbf{1}_d$, where $c$ is any positive constant.*

*Remark* 3.2. The above theorem indicates that the minimizer of $\mathbf{I}_{p,q}(\boldsymbol{w})$ has equal components. This minimizer is unique up to a scalar factor $c$.

**Theorem 3.3.** *For $p = 1$ and $q = 2$,*

$$\left\| \frac{\boldsymbol{w}}{\|\boldsymbol{w}\|_2} - d^{-\frac{1}{2}} \cdot \mathbf{1}_d \right\|_2 = \sqrt{2\mathbf{I}_{1,2}(\boldsymbol{w})}.$$

*Remark* 3.3. The above theorem shows that $\mathbf{I}_{1,2}(w)$ quantifies the distance between the $\boldsymbol{w}$, scaled to have unit $l_2$ norm, and the unit vector with equal components. Thus, as $\mathbf{I}_{1,2}(w)$ decreases, the normalized $\boldsymbol{w}$ approaches $d^{-\frac{1}{2}} \cdot \mathbf{1}_d$.

**Theorem 3.4.** *Let $p = 1$, $q = 2$. Assume that one component of $\boldsymbol{w}$ is strictly larger than the others. We remove $\tilde{c}$ ($0 < \tilde{c} < w_1$) from that component and add $\tilde{c}/(d-1)$ to the remaining components and denote the resulting vector by $\tilde{\boldsymbol{w}}$. Without loss of generality, suppose $w_1 = w_{max}$ and $w_1 > w_i$ ($i = 2, \ldots, d$). Then,*

$$\tilde{\boldsymbol{w}} = [w_1 - \tilde{c}, w_2 + \tilde{c}/(d-1), \ldots, w_d + \tilde{c}/(d-1)].$$

*If $\tilde{w}_1 = \tilde{w}_{max}$, we have*

$$\mathbf{I}_{1,2}(\tilde{\boldsymbol{w}}) < \mathbf{I}_{1,2}(\boldsymbol{w}).$$

*Remark* 3.4. Ideally, a sparsity metric should decrease if we remove part of the largest component and distribute its value to the remaining components, while ensuring that the largest component remains the largest. This aligns with the property of PQ Index stated in the above theorem. Since $\tilde{w}_1 - \tilde{w}_i < w_1 - w_i$ and $\tilde{w}_i - \tilde{w}_j = w_i - w_j$ ($i, j \in \{2, \ldots, d\}$), the Gini Index and the Maximum Pairwise Difference also have the above property.

### 3.2 A COMPARISON AMONG THE SPARSITY MEASURES

First, we show the connection among the PQ Index, the Gini Index, and the Maximum Pairwise Difference by the following theorem.

**Theorem 3.5.** *Let $p = 1$ and $q = 2$. We have*

$$Gini(\boldsymbol{w}) \leq \frac{d}{2\|\boldsymbol{w}\|_2} MPD(\boldsymbol{w}), \tag{1}$$

$$MPD(\boldsymbol{w}) \leq 2\|\boldsymbol{w}\|_2 \sqrt{2\mathbf{I}_{1,2}(\boldsymbol{w})}, \tag{2}$$

$$\mathbf{I}_{1,2}(\boldsymbol{w}) \leq Gini(\boldsymbol{w}). \tag{3}$$

*Remark* 3.5. The above theorem shows that for vectors $\boldsymbol{w}$ with a fixed $\|\boldsymbol{w}\|_2$, any one sparsity measure can be bounded in terms of the others. We need to fix $\|\boldsymbol{w}\|_2$ since $MPD(\boldsymbol{w})$ is not scale-invariant. Bounding the PQ Index by the Gini Index, or vice versa, does not impose this requirement.

Second, we analyze the differences between the Maximum Pairwise Difference, PQ Index, and Gini Index. The main difference between the Maximum Pairwise Difference and the latter two is that $MPD(\boldsymbol{w})$ depends on the largest and the smallest components of $\boldsymbol{w}$, whereas $\mathbf{I}_{p,q}(\boldsymbol{w})$ and $Gini(\boldsymbol{w})$ also depend on the values of the other components. Specifically, the following theorem holds for PQ Index:

**Theorem 3.6.** *Denote $\dot{\boldsymbol{w}}$ as the $(d-2)$-dimensional vector obtained by removing the components of $\boldsymbol{w}$ that are equal to $w_{max}$ or $w_{min}$. Let $\boldsymbol{w}^{(1)}$ and $\boldsymbol{w}^{(2)}$ be two d-dimensional vectors with the same number of largest components and the same number of smallest components. If*

$$\mathbf{I}_{p,q}(\boldsymbol{w}^{(1)}) < \mathbf{I}_{p,q}(\boldsymbol{w}^{(2)}),$$

$$w_{max}^{(1)}/\|\boldsymbol{w}^{(1)}\|_q = w_{max}^{(2)}/\|\boldsymbol{w}^{(2)}\|_q,$$

$$w_{min}^{(1)}/\|\boldsymbol{w}^{(1)}\|_q = w_{min}^{(2)}/\|\boldsymbol{w}^{(2)}\|_q,$$

*we also have $\mathbf{I}_{p,q}(\dot{\boldsymbol{w}}^{(1)}) < \mathbf{I}_{p,q}(\dot{\boldsymbol{w}}^{(2)})$.*

*Remark* 3.6. The above theorem shows that for two unit vectors with the same largest and smallest components, their relative sparsity, as measured by PQ Index, is determined by the values of their remaining components after removing the largest and smallest ones. In contrast, for two such vectors, their Maximum Pairwise Difference is always the same.

We also highlight the differences between the PQ Index and the Gini Index. We restrict $\boldsymbol{w}$ such that $\|\boldsymbol{w}\|_1 = 1$. When $w_1 \geq \cdots \geq w_d$, it can be shown that

$$Gini(\boldsymbol{w}) = \frac{1}{d} \sum_{i=1}^{d} (d + 1 - 2i) w_i.$$

For a different ordering of the components of $\boldsymbol{w}$, we replace $(w_1, \ldots, w_d)^\top$ in the above formula with $\boldsymbol{w}$ after rearranging its components in decreasing order. Therefore, given that $\|\boldsymbol{w}\|_1 = 1$, $Gini(\boldsymbol{w})$ is a piece-wise linear function of $\boldsymbol{w}$. In contrast, $\mathbf{I}_{p,q}(\boldsymbol{w})$ is a smooth function. We visualize $Gini(\boldsymbol{w})$ and $\mathbf{I}_{1,2}(\boldsymbol{w})$ in Figure 1 for $d = 3$. As highlighted in Remark 3.1 and 3.2, when the sparsity of the vector reaches its maximum (i.e., a one-hot vector), the MPD also attains its maximum. Conversely, a uniform vector minimizes both sparsity and MPD. These relationships illustrate how sparsity reflects group-level disparities, thereby supporting its role as a fairness-relevant measure.

## 4 UNIFYING GROUP FAIRNESS WITH SPARSITY

In this section, we formulate a unified fairness framework based on the idea that sparsity is the inverse of fairness. In general, we replace the Maximum Pairwise Difference used in existing fairness metrics with a sparsity measure over $w$, where the length of the vector $w$ equals the number of sensitive groups in the input. Denote the input vector as $X \in \mathcal{X}$, the target vector as $Y \in \mathcal{Y}$, and the sensitive attribute vector as $A \in \mathcal{A}$, where $A$ may or may not be a subset of $X$. Let $X_a$ and $Y_a$ be the data points belonging to a subgroup $a \in \mathcal{A}$. Let $|\mathcal{Y}|$ and $|\mathcal{A}|$ be the cardinalities of $Y$ and $A$, respectively. For a function $f : X \mapsto f(X)$, let $S : f(X) \mapsto S(f(X))$ be any sparsity measure imposed on $f$, and $g : f(X) \mapsto g(f(X))$ be a model performance evaluation metric based on, e.g., the *Confusion Matrix* for classification or *Mean Squared Error* for regression.

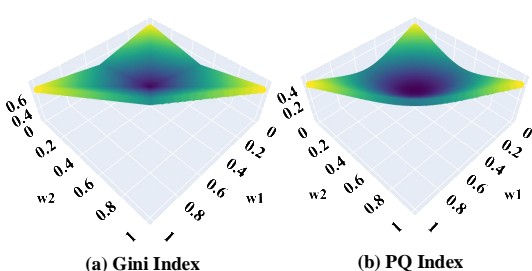

**(a) Gini Index**   **(b) PQ Index**

Figure 1: The plots of (a) $Gini(w)$ and (b) $\mathbf{I}_{1,2}(w)$ for $d = 3$ and $\|w\|_1 = 1$. In each plot, the horizontal axes correspond to $w_1$ and $w_2$ (where $w_3 = 1 - w_1 - w_2$). The vertical axis shows the value of Gini Index or PQ Index. Since there are 6 possible permutations of $[w_1, w_2, w_3]$, $Gini(w)$ is composed of subsets of 6 distinct planes. In contrast, $\mathbf{I}_{1,2}(w)$ has a smooth surface. Both Gini Index and PQ Index attain their minimum at $w = [3^{-1}, 3^{-1}, 3^{-1}]^{\mathsf{T}}$.

We denote the sparsity based metrics in the form $S\text{-}*$, where "$*$" is a placeholder for an existing fairness criterion, such as statistical parity. Let $a$ be a vector with components $a_i \in \mathcal{A}$ ($i = 1, \ldots, |\mathcal{A}|$). Let $m_i \in \mathbb{R}$ represent outputs from a function that depends on the index $i$. Specifically, $m_i$ can be $f(X_{a_i})$ or $g(f(X_{a_i}), Y_{a_i})$. Define $[m_i]_{i=1}^{|\mathcal{A}|} \triangleq [m_1, \ldots, m_{|\mathcal{A}|}]^{\mathsf{T}}$. Then, $S\text{-}*$ can be expressed as $S([m_i]_{i=1}^{|\mathcal{A}|})$. We summarize the criteria discussed in this paper in Table 1.

Table 1: Group fairness criteria discussed in the main text. For classification, we focus on Statistical Parity and Equalized Odds. For regression, we use Statistical Parity based on Kolmogorov-Smirnov (KS) distance and propose EO definition in MPD form. We then reformulated these criteria incorporating sparsity measure $S(\cdot)$. Fairness criteria proposed in this work are highlighted in gray, with details in Section 4. A complete table of criteria can be found in Appendix B.3

| Problem | Criteria | Expression |
|---------|----------|------------|
| Classification | Statistical Parity | $\max_{y \in \mathcal{Y}} \max_{a,a' \in \mathcal{A}} \left\| \mathbb{E}(f(X_a) = y) - \mathbb{E}(f(X_{a'}) = y) \right\|$ |
| | $S$-Statistical Parity | $\max_{y \in \mathcal{Y}} S\left( \mathbb{E}(f(X_{a_i}) = y)_{i=1}^{|\mathcal{A}|} \right)$ |
| | Equalized Odds | $\max_{y,y' \in \mathcal{Y}} \max_{a,a' \in \mathcal{A}} \left\| \mathbb{P}_{y',a}(f(X) = y) - \mathbb{P}_{y',a'}(f(X) = y) \right\|$ |
| | $S$-Equalized Odds | $\max_{y \in \mathcal{Y}} S\left( [g(f(X_{a_i}), Y_{a_i})]_{i=1}^{|\mathcal{A}|} \right)$ |
| Regression | Statistical Parity | $\sup_{y \in \mathcal{Y}} \max_{a,a' \in \mathcal{A}} \left\| \mathbb{P}_a(f(X) \leq y) - \mathbb{P}_{a'}(f(X) \leq y) \right\|$ |
| | $S$-Statistical Parity | $\sup_{y \in \mathcal{Y}} S\left( [\mathbb{P}_{a_i}(f(X) \leq y)]_{i=1}^{|\mathcal{A}|} \right)$ |
| | Equalized Odds | $\max_{a,a' \in \mathcal{A}} \left\| g(f(X_a), Y_a) - g(f(X_{a'}), Y_{a'}) \right\|$ |
| | $S$-Equalized Odds | $S\left( [g(f(X_{a_i}), Y_{a_i})]_{i=1}^{|\mathcal{A}|} \right)$ |

### 4.1 STATISTICAL PARITY (SP)

Statistical Parity (*a.k.a. Demographic Parity*) assesses whether the predicted outcome of a model is independent of sensitive attributes (e.g., race, gender, or education). Enforcing statistical parity ensures that the likelihood of a specific model outcome is equal across different sensitive groups, regardless of group membership.

**Classification.** Statistical Parity has been used extensively in classification problems to quantify algorithmic fairness for classification Calders et al. (2009). Although its oracle form is proposed for binary classification problems, recent work Alghamdi et al. (2022); Xian et al. (2023); Denis et al. (2024) has advanced its usage to multi-classification problems.

**Definition 4.1** (Statistical Parity (*classification*))**.** A classifier $f: \mathcal{X} \to \mathcal{Y}$ satisfies SP if the following quantity is equal to 0:

$$\max_{y \in \mathcal{Y}} \max_{a,a' \in \mathcal{A}} \left| \mathbb{E}(f(X_a) = y) - \mathbb{E}(f(X_{a'}) = y) \right|.$$

The Maximum Pairwise Difference (Definition 3.1) is calculated among group-wise outputs, and then the maximum value is taken over all classes.

**Definition 4.2** ($S_c$-Statistical Parity)**.** The sparsity-based statistical parity is measured by

$$\max_{y \in \mathcal{Y}} S\left([\mathbb{E}(f(X_{a_i}) = y)]_{i=1}^{|\mathcal{A}|}\right).$$

If $S(\cdot)$ in the above expression is the Maximum Pairwise Difference, the classifier reduces to Definition 4.1. The suffix "c" stands for classification. One may also consider replacing the `max` operation over multiple classes with other measures, such as `mean` or `sum`. (See Appendix D.1)

**Regression.**    In the context of fair regression, the following definition of (strong) statistical parity has been used frequently in the literature (Agarwal et al., 2019; Jiang et al., 2020; Silvia et al., 2020; Chzhen et al., 2020). For regression models, we assume that $\mathcal{Y} \subseteq \mathbb{R}$.

**Definition 4.3** (Statistical Parity (*regression*))**.** A regression model $f: \mathcal{X} \to \mathcal{Y}$ is considered to satisfy SP if the following quantity is equal to 0:

$$\sup_{y \in \mathcal{Y}} \max_{a,a' \in \mathcal{A}} \left| \mathbb{P}_a(f(X) \leq y) - \mathbb{P}_{a'}(f(X) \leq y) \right|,$$

where $\mathbb{P}_a(\cdot)$ denotes the probability conditional on $A = a$. Here, the difference between $\mathbb{P}_a(f(X) \leq y)$ and $\mathbb{P}_{a'}(f(X) \leq y)$ is measured using the Kolmogorov-Smirnov distance (Lehmann & Romano, 2006).

The above is considered a stronger fairness criterion than general statistical parity, since it accounts for the entire shape of the distribution (Silvia et al., 2020), ensuring that the distributions remain similar across different groups.

**Definition 4.4** ($S_r$-Statistical Parity)**.** The sparsity-based statistical parity is measured by

$$\sup_{y \in \mathcal{Y}} S\left([\mathbb{P}_{a_i}(f(X) \leq y)]_{i=1}^{|\mathcal{A}|}\right) \tag{4}$$

This definition borrows the idea from the Kolmogorov-Smirnov (KS) distance and finds the maximum sparsity among group CDFs. If $S(\cdot)$ is taken to be the Maximum Pairwise Difference, the above expression reduces to Definition 4.3.

In practice, the closed form of each conditional CDF is often unknown. To address this issue, we may approximate them with empirical cumulative distribution functions.

## 4.2 Equalized Odds (EO)

Hardt et al. (2016) introduced the concept of Equalized Odds (EO) that incorporates the distribution of the ground truth label by enforcing $f(X) \perp A \mid Y$, where $\perp$ denotes the independence between two random variables. Consequently, when the sensitive attribute $A$ is related to the ground truth label $Y$, EO requires that the predictions $f(X)$ reveal no additional information about $A$ beyond what is already contained in $Y$ (Woodworth et al., 2017).

**Classification.**    In classification, EO measures fairness from a different perspective. Compared with SP, it has the following two key differences (Dwork et al., 2012; Agarwal et al., 2018): **1)** A classifier can achieve a low SP score by matching $\mathbb{P}(f(X) = 1)$ across groups, even if it makes accurate predictions for the majority group of $A$ while producing random predictions for the others. **2)** A perfect classifier may violate SP if $Y$ is dependent on $A$.

**Definition 4.5** (Equalized Odds)**.** For a classifier $f: \mathcal{X} \to \mathcal{Y}$, equalized odds (Hardt et al., 2016; Xian & Zhao, 2024) considers

$$\max_{y,y' \in \mathcal{Y}} \max_{a,a' \in \mathcal{A}} \left| \mathbb{P}_{y',a}(f(X) = y) - \mathbb{P}_{y',a'}(f(X) = y) \right|,$$

where $\mathbb{P}_{y,a}(f(X) = y)$ denotes $\mathbb{P}(f(X) = y \mid Y = y, A = a)$. In multi-class scenarios, $y$ and $y'$ are vectors of class labels.

Next, we propose a more general definition for EO by incorporating sparsity measures. Let $g : (Y, f(X)) \mapsto g(Y, f(X)) \in \mathbb{R}$ be an arbitrary model performance evaluation metric. For example, $g(\cdot)$ can be the accuracy of the model or the model loss such as Cross Entropy (CE) Loss.

**Definition 4.6** ($S_c$-Equalized Odds). The sparsity-based equalized odds for classifiers considers:

$$\max_{y \in \mathcal{Y}} S\big([g(f(X_{a_i}), Y_{a_i})]_{i=1}^{|\mathcal{A}|}\big). \tag{5}$$

In multi-class classification, we evaluate equation 5 for each class separately and take the maximum value. Following Definition 4.5 and previous work Alghamdi et al. (2022), we define $g(\cdot)$ as the average of the True Positive Rate (TPR) and False Positive Rate (FPR).

**Regression.**   To the best of our knowledge, there is no existing fairness criterion similar to the formulation of EO in regression problems. We define EO in regression as follows for completeness:

**Definition 4.7** (Equalized Odds (*regression*)). For regression data $(X_{a_i}, Y_{a_i})$ in each group, the EO can be expressed as

$$\max_{a, a' \in \mathcal{A}} \Big| g(f(X_a), Y_a) - g(f(X_{a'}), Y_{a'}) \Big|.$$

This naturally extends to $S$-EO by incorporating sparsity measures.

**Definition 4.8** ($S_r$-Equalized Odds). The sparsity-based equalized odds for regression models considers

$$S\big([g(f(X_{a_i}), Y_{a_i})]_{i=1}^{|\mathcal{A}|}\big),$$

where $g : (Y, f(X)) \mapsto g(Y, f(X)) \in \mathbb{R}$ is model performance evaluation metric for regression models. The function $g(\cdot)$ can be the *Mean Squared Error (MSE)* or, when additional information about the distribution of $Y$ is available, the log-likelihood.

## 5 EXPERIMENTS

### 5.1 EXPERIMENTAL SETUP

In this section, we conduct experiments to validate our proposed criteria in comparisons with other established fairness notions and evaluate them across different bias mitigation algorithms. We aim to address the following research questions:

- **Q1:** Do sparsity-based metrics align with MPD-based metrics across different benchmarks?
- **Q2:** In which scenarios do the two evaluation frameworks exhibit divergent behaviors?

We apply the PQ Index ($p = 1, q = 2$) (Diao et al., 2023) as the sparsity measure $S(\cdot)$ for all the primary results. On each of the problem and dataset, we evaluate bias mitigation algorithms using our proposed criteria and compare the results against the existing criteria. Following previous practice (Agarwal et al., 2018; Wei et al., 2021; Alghamdi et al., 2022), we include the sensitive attribute $A$ in the input $X$ for consistent comparisons in all of our experiments, except for the simulated data in the regression setting. The detailed configurations are provided in Appendix C.2.

**Datasets.**   For the classification task, we follow prior work in fair classification (Agarwal et al., 2018; Cho et al., 2020b; Jeong et al., 2022; Alghamdi et al., 2022; Xian et al., 2023) and include datasets such as *UCI Adult*, *COMPAS*, *HSLS*, *ACSIncome*, and *Enem*. And for regression, we consider *Communities & Crimes* and *LawSchool*, which have been commonly used as benchmarks in fair regression studies (Agarwal et al., 2019; Chzhen et al., 2020; Xian et al., 2024). Details on each dataset can be found in Appendix C.1.

**Baselines.**   For the **classification** setting, we include the following bias mitigation algorithms: *Reweight* (Kamiran & Calders, 2012), *FairRR* (Zeng et al., 2024), *Reduction* (Agarwal et al., 2018), *Rejection* (Kamiran et al., 2012), *EqOdds* (Hardt et al., 2016), *CalEqOdds* (Pleiss et al., 2017), *FairProj* (Alghamdi et al., 2022) and *LinearPost* (Xian et al., 2023). Among those existing algorithms, only *FairProj* and *LinearPost* are capable of handling multi-classification problem. For *FairProj*, we test both Kullback–Leibler divergence (KL) and Cross Entropy (CE) as the divergence measure used in the algorithm.

In contrast to classification, fairness in **regression** has received relatively less attention. In our benchmark experiments, we evaluate three representative bias mitigation algorithms for regression:

*FairReg* (Agarwal et al., 2019), *WassBC* (Chzhen et al., 2020), *LinearPost* (Xian et al., 2024). We include details on the hyperparameter selections in Appendix C.2.

Most of these algorithms belong to the post-processing category, except for *Reduction* and *Fair-Reg*, which are in-processing methods. A logistic regression model is used as the base model for classification benchmarks, while a linear regression model is used for regression.

## 5.2 ALIGNMENT WITH EXISTING METRIC

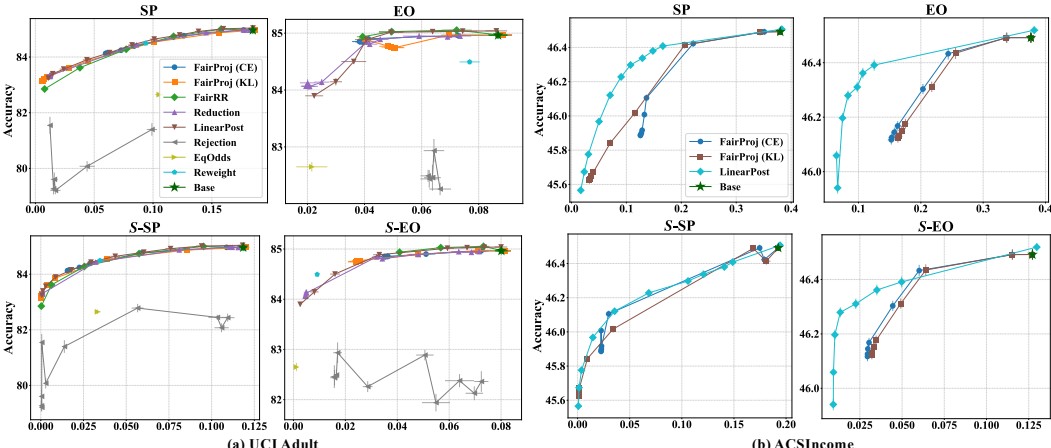

Figure 2: Comparison of sparsity criteria with baseline criteria in two classification dataset. The top row shows results from baseline criteria; the bottom row shows results from the proposed sparsity criteria. The x-axis of each plot represents the value of various criteria.

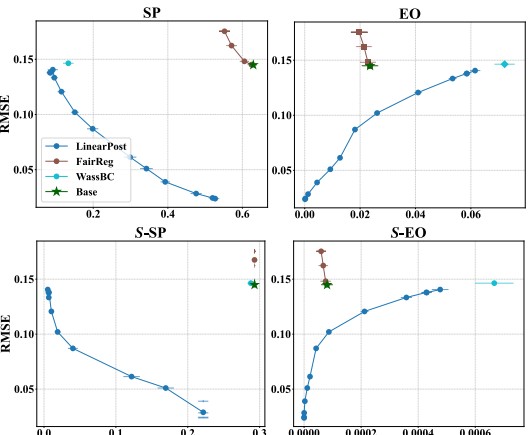

Figure 3: Comparison in *Community & Crimes*.

Figure 2 illustrates the comparison between the proposed fairness criteria ($S$-$*$) and their existing counterparts for two classification datasets: *UCI Adult* ($|\mathcal{Y}| = 2$, $|\mathcal{A}| = 2$) and *ACSIncome* ($|\mathcal{Y}| = 5$, $|\mathcal{A}| = 5$). For methods that generate fairness-accuracy trade-off curves, we select a range of fairness budgets corresponding to the respective fairness criteria to illustrate their effects. The results reveal similar trends in the trade-off curves between the $S$-$*$ metrics and existing fairness metrics across the benchmark methods. These observations hold for both binary-class/binary-group problems and multi-class/multi-group problems.

Furthermore, the proposed notions do not alter the trade-off patterns. For instance, the *Linear-Post* algorithm achieves the best trade-off curves for SP and EO, and the same trend is observed for $S$-SP and $S$-EO. These findings suggest that in classification tasks, sparsity-based metrics effectively capture the characteristics of the original fairness criteria while ensuring equal consideration for all groups within the sensitive attribute. Consequently, $S$-$*$ represent a valuable alternative for measuring fairness in classification problems.

In Figure 3, we demonstrate similar comparisons for regression dataset *Communities & Crime* ($|\mathcal{A}| = 2$). In this data set, we observe that $S$-SP and $S$-EO resemble the trade-offs of the benchmark algorithms compared to the MPD versions. Across experiments, we see that *FairReg*, as an in-processing reduction algorithm, always reduces to the unconstrained case. Instead, *LinearPost* and *WassBC* directly search for the Bayes Optimal regressor under the fairness constraint through post-processing and disregard the unconstrained model fit. While these methods are not deliberately designed for mitigating EO violations in regression, *LinearPost* successfully produces points achieving the desired *RMSE* and $S$-EO values. Recall in Section 3 that the elements in $\boldsymbol{w}$ are required to be non-negative. However, quantities passed into the sparsity measure may contain negative, zero, or extreme small values, which cause instability in the fairness criteria. We adopt an exponential

transformation to enforce the positivity and show the result in $S$-EO in Figure 3. We further examine the exponential transformation under different settings in Appendix D.1.

### 5.3 WHEN AND HOW THE TWO PARADIGMS DIFFER

**Intersectional Fairness** Intersectional fairness considers multiple sensitive attributes at the same time (Crenshaw, 2013; Gaucher et al., 2023), whereas most prior research on group fairness has focused on a single dimension of group identity (Yang et al., 2020). In this section, we target a common scenario in intersectional fairness where the number of sensitive groups becomes large. Using both simulated data and real-world example (*Adult*), we observe similarities and differences for MPD and $S$-∗ metrics. For the simulated binary classification dataset, we interpolate the class weight by adjusting the available training samples for each group and fix the maximum class weight difference as the group size increases. To achieve large sensitive groups in the *Adult* dataset, we utilize intersections of gender, race and descretized age. (See Appendix C.1). Experiments are conducted with three random data splits.

For the unconstrained model (*Base*), we observe that the MPD-based SP remains at the same level regardless of the group size in the simulated setting, since it only reflects the maximum group disparity. In contrast, $S$-SP captures the addition of groups with class imbalances smaller than the maximum difference. As the group size increases, $S$-SP from the *Base* model decreases due to the presence of more intermediate groups, which dilutes the class distribution across the dataset and makes it less sparse. The results align with the findings of Theorem 3.6, which states the relative sparsity of two vectors is determined by the rest of the components if they have the same maximum and minimum components.

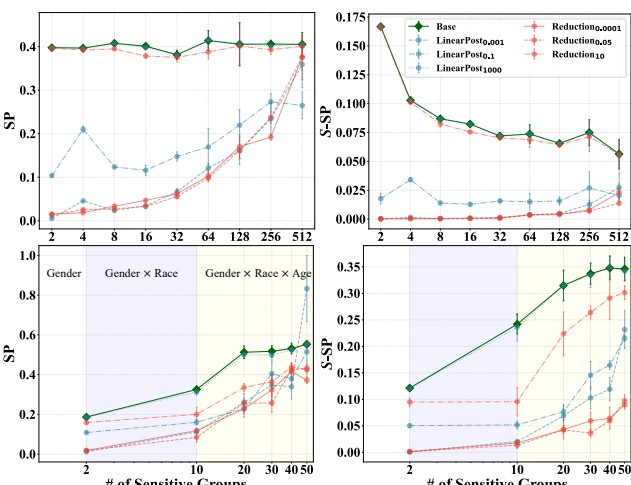

Figure 4: **Top**: *Simulated* binary classfication dataset. **Bottom**: *Adult* dataset. Legend indicates bias mitigation methods, with subscripts denoting hyperparameter.

Furthermore, in the simulated setting, we observe that existing bias mitigation algorithms exhibit inconsistent debiasing performance for SP as the group size increases, while they remain effective for $S$-SP across various sensitive group sizes. This discrepancy appears to stem from the algorithms successfully balancing predictions for most groups but failing for a few under large number of groups. It leads to substantial effect on SP but only a minimal impact on the sparsity-based measure.

In the multi-group *Adult* dataset results, we observe that both the SP and the $S$-SP values increase as the grouping granularity increases. Specifically, when the number of groups becomes large, edge cases may arise where one class is entirely absent within certain groups. In such cases, SP can produce extreme values (e.g., the result of *LinearPost*$_{0.001}$ at a group size of 50), whereas $S$-SP provides a more stable evaluation by incorporating group distribution through sparsity. Across both experiments, we demonstrate that the sparsity-based metric captures subtle group disparities overlooked by MPD and exhibits greater robustness under severe class imbalance within groups.

**Fair Recommendation System** As highly data-driven systems, recommendation systems (RS) are susceptible to data and algorithmic biases that can lead to unfair outcomes. Ensuring fairness in recommendation systems remains an open challenge and requires multifaceted evaluation approaches (Li et al., 2023b; Ge et al., 2024; Zhao et al., 2025). In this section, we consider a simplified online setting where we evaluate the recommendation quality for all active users.

We use the *MovieLens* dataset (Harper & Konstan, 2015) and filter out users and movies with fewer than 20 ratings. The resulting dataset contains 67,898 ratings from 610 users. We split the data into training and test sets using an 80/20 ratio at the user–movie level. A Matrix Factorization model (Koren et al., 2009) is then trained on the training set to perform rating prediction.

For fairness evaluation, we begin by randomly sampling $k = 10$ users and adding them to the test set. Subsequently, additional uniformly sampled batches of $k$ new users arrive sequentially, incrementally expanding the test set until all users are included. For each expanded test set, we compute the RMSE of movie rating predictions for each user and derive group fairness metrics using MPD, the Gini Index, and the PQ Index, treating each user as an individual group. We run each evaluation with 5 independent random repeats.

Figure 5 presents the trajectories of these metrics as the number of active users increases. We observe that the MPD metric increases monotonically, whereas sparsity-based metrics remain relatively stable with only minor fluctuations. the results show that the MPD-based metric is not only highly sensitive to outliers introduced by new samples but can also become dominated by them, whereas the sparsity-based metric remains comparatively stable. Such stability is theoretically expected, given that new samples are drawn uniformly and the pretrained model remains unchanged.

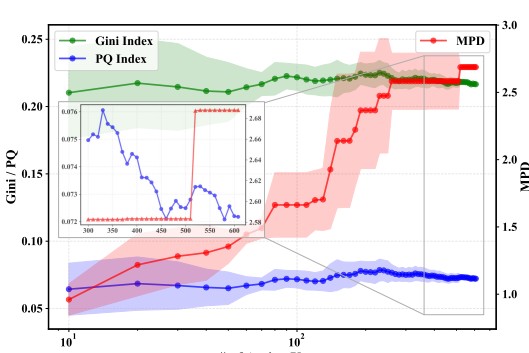

A closer inspection of the tail region of the curves reveals that even when MPD reaches a plateau, sparsity-based metrics can still increase or decrease. This is because sparsity measures

Figure 5: Comparisons of three evaluation metrics for a Recommendation System in an online setting. Recommendation quality is assessed using RMSE.

depend on intermediate components of the input vector rather than solely on the extreme values (Theorem 3.6). Finally, we note that sparsity-based metrics exhibit convergence as the number of input groups grows.

**Discussion** We explicitly examine conditions under which sparsity-based metrics diverge from MPD-based metrics. The three representative cases are summarized below:

1. **Fixed total samples and Fixed maximum disparity.** When the maximum gap is held constant, sparsity-based metrics capture changes arising from the intermediate group values (Remark 3.1 and Remark 3.2). In contrast, MPD remains unchanged because it depends solely on the largest pairwise disparity.

2. **Fixed total samples and Varying group disparities.** In this setting, both sparsity-based and MPD-based metrics increase as group granularity becomes finer. However, overly fine-grained grouping can induce extreme class imbalance in model predictions (e.g., certain classes disappearing). Since sparsity-based metrics dilute the influence of any single group, they exhibit greater robustness under such conditions.

3. **Varying total samples and Varying group disparities.** When new groups (samples) are added, MPD is dominated by the two groups with the largest disparity and cannot recover as long as these extreme groups remain present. In contrast, sparsity-based metrics demonstrate improved stability and convergence as additional groups are introduced.

Across all three cases, the divergence between the two metrics becomes more pronounced when the number of groups is large. If the practitioner is primarily concerned with the worst-case fairness violation, MPD is the appropriate choice. Otherwise, sparsity-based metrics provide a viable alternative with stronger stability and robustness.

## 6 CONCLUSION

In this paper, we propose to unify various fairness criteria in machine learning with a sparsity measure. We highlight that sparsity, inherently designed as a measure of inequality, can also serve as a viable definition of fairness. Our work provides deeper insight into the properties of the PQ Index and a theoretical comparison of MPD, the Gini Index, and the PQ Index. Building on this foundation, we propose a unified framework that incorporates sparsity measures into fairness criteria such as statistical parity (SP) and equalized odds (EO). Through comprehensive benchmarking across multiple datasets and bias mitigation algorithms, we demonstrate that the proposed framework aligns well with the state-of-the-art approaches. Future research directions include developing fair algorithms that utilize PQI or other sparsity measures for bias mitigation.

## ETHICS STATEMENT

We acknowledge several important ethical concerns related to fairness research in machine learning.

**Dataset limitations**  We recognize the limitations of commonly used benchmark datasets, may include outdated labels ( i.e. income information in Adult), inherent biases in data collection (racial bias in COMPAS), which limits their ability to fully reflect real-world decision-making contexts. While such datasets are widely used for comparative analysis, we caution against interpreting empirical results in isolation from these known limitations. Throughout this paper, we focus on illustrating the behavior of different fairness metrics, rather than promoting specific deployment-ready solutions.

**Broader society impact**  Beyond technical contributions, our work has potential societal impact by promoting fairness measures that are more aligned with social equity principles. By connecting perspectives from the social sciences with algorithmic fairness, we aim to support the development of more inclusive and responsible AI systems that better serve diverse populations. However, we are aware that sparsity-based fairness measures, like any fairness criterion, must be applied with context. In particular, optimizing for sparsity may carry the risk of underrepresenting smaller or marginalized subgroups if applied without careful consideration. Our framework is designed to surface structural relationships between fairness metrics, not to prescribe a universal approach. There is a risk that over-reliance on a single measure, such as sparsity, may oversimplify complex social dynamics or overlook harms not captured by the chosen formalism. We emphasize that any fairness intervention should be guided by domain knowledge, stakeholder input, and sensitivity to the social and legal implications in deployment scenarios.

**Access to sensitive features**  Our experimental framework assumes access to demographic attributes (e.g., race, gender) for the purpose of fairness evaluation. In real-world systems, however, such sensitive attributes are often unavailable due to legal restrictions, data privacy concerns, or lack of user consent. We acknowledge that collecting and using such data requires careful attention to regulatory frameworks (e.g., GDPR), informed consent, and community norms. We do not advocate for indiscriminate collection of sensitive attributes, and we recognize the risk that their misuse can exacerbate harms rather than mitigate them. In settings where sensitive attributes are not accessible, fairness evaluation may need to rely on proxy features, post-hoc audits, or participatory methods involving stakeholders. We encourage future work to explore fairness-aware learning under limited or uncertain demographic information, and engage with communities impacted by algorithmic decision-making.

## REPRODUCIBILITY STATEMENT

We provide the complete source code in the supplementary materials. Further details on the theoretical analysis, experimental setup including hyperparameters and datasets, and results are documented in the Appendix.

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

# Appendix

## A Discussion

### A.1 Limitations

Although our theoretical analysis and empirical results suggest that sparsity possesses desirable properties for measuring fairness in machine learning and aligns well with current algorithmic fairness research, several limitations warrant further discussion.

First, while our work primarily focuses on the technical alignment between fairness and sparsity, its broader applicability to AI or social fairness remains to be explored. A more comprehensive evaluation is needed to identify practical scenarios where the $S\text{-}*$ metric may be better suited than MPD-based metrics.

Second, as noted in the main text, sparsity-based metrics may introduce numerical instability compared to MPD-based measures. While we employ an exponential transformation to mitigate this issue, alternatives beyond heuristic approaches require further investigation.

### A.2 Broader Impact

This work draws inspiration from the Gini Index, a well-established measure in the social sciences, to bridge algorithmic fairness with broader notions of equity observed under real-world contexts. By grounding our approach in sparsity, we offer a norm-based fairness evaluation framework that is not only interpretable but also directly optimizable. Unlike previous approaches that rely on indirect surrogates such as mutual information due to the non-optimizable nature of MPD, our formulation enables straightforward integration of norm-based regularization into learning objectives.

### A.3 Use of Large Language Models

In this work, we used large language models (LLMs) to assist with manuscript editing. LLMs were used to help polish the language of the manuscript. This includes surface-level edits such as improving clarity, grammar, and conciseness of English expressions. All technical content, algorithmic designs, and empirical results were authored and validated by the authors. No part of the scientific contributions was generated by or delegated to an LLM.

## B Theoretical Analysis

### B.1 Ideal Properties for Sparsity Measures

Hurley & Rickard (2009) have outlined the following desirable properties for sparsity measures, which were originally introduced in the works of Dalton (1920); Rickard & Fallon (2004):

**(D1)** Robin Hood: For any $w_i > w_j$ $(i, j \in \{1, \ldots, d\})$ and $\alpha \in (0, (w_i - w_j)/2)$, we have
$$S\left([w_1, \ldots, w_i - \alpha, \ldots, w_j + \alpha, \ldots, w_d]^{\mathrm{T}}\right) < S(\boldsymbol{w}).$$

**(D2)** Scaling: $S(\alpha\boldsymbol{w}) = S(\boldsymbol{w})$ for any $\alpha > 0$.

**(D3)** Rising Tide: $S(\boldsymbol{w} + \alpha) < S(\boldsymbol{w})$ for any $\alpha > 0$ and $w_i$ not all the same.

**(D4)** Cloning: $S(\boldsymbol{w}) = S([\boldsymbol{w}^{\mathrm{T}}, \boldsymbol{w}^{\mathrm{T}}]^{\mathrm{T}})$.

**(P1)** Bill Gates: For any $i = 1, \ldots, d$, there exists $\beta_i > 0$ such that for any $\alpha > 0$ we have
$$S\left([w_1, \ldots, w_i + \beta_i + \alpha, \ldots, w_d]^{\mathrm{T}}\right) > S\left([w_1, \ldots, w_i + \beta_i, \ldots, w_d]^{\mathrm{T}}\right).$$

**(P2)** Babies: $S\left([\boldsymbol{w}^{\mathrm{T}}, 0]^{\mathrm{T}}\right) > S(\boldsymbol{w})$ for any non-zero $\boldsymbol{w}$.

Hurley & Rickard (2009) showed that the Gini Index satisfies the aforementioned six properties, and Diao et al. (2023) proved that the same holds for the PQ Index as well. For the Maximum Pairwise Difference, only properties **(D4)** and **(P1)** are satisfied. The above results are summarized in Table 2, and the explanations for the Maximum Pairwise Difference are as follows:

1. For **(D1)**, when $w_i < w_{max}$ and $w_j > w_{min}$ (recall that $w_{max}$ and $w_{min}$ are the values of the largest and smallest components of $\boldsymbol{w}$, respectively), we have

$$MPD\left([w_1, \ldots, w_i - \alpha, \ldots, w_j + \alpha, \ldots, w_d]^{\mathrm{T}}\right) = MPD(\boldsymbol{w}).$$

   Therefore, Robin Hood does not hold for the Maximum Pairwise Difference.

2. For **(D2)**,

$$MPD(\alpha\boldsymbol{w}) = \alpha MPD(\boldsymbol{w}), \ \forall \alpha > 0.$$

   Therefore, Scaling does not hold for the Maximum Pairwise Difference.

3. For **(D3)**,

$$MPD(\boldsymbol{w} + \alpha) = \max_{i,j\in\{1,\ldots,d\}} |w_i + \alpha - w_j - \alpha| = MPD(\boldsymbol{w}), \ \forall \alpha > 0.$$

   Therefore, Rising Tide does not hold for the Maximum Pairwise Difference.

4. For **(D4)**, since the largest components of $\boldsymbol{w}$ equals the ones of $[\boldsymbol{w}^{\mathrm{T}}, \boldsymbol{w}^{\mathrm{T}}]^{\mathrm{T}}$, and the same holds for their smallest components,

$$MPD(\boldsymbol{w}) = MPD([\boldsymbol{w}^{\mathrm{T}}, \boldsymbol{w}^{\mathrm{T}}]^{\mathrm{T}}).$$

   Therefore, Cloning holds for the Maximum Pairwise Difference.

5. For **(P1)**, we may take $\beta_i = w_{max} - w_i + 1$. Then,

$$MPD\left([w_1, \ldots, w_i + \beta_i + \alpha, \ldots, w_d]^{\mathrm{T}}\right) = \alpha + 1 + w_{max} - w_{min}.$$

   Since

$$MPD\left([w_1, \ldots, w_i + \beta_i, \ldots, w_d]^{\mathrm{T}}\right) = 1 + w_{max} - w_{min},$$

   we have that Bill Gates holds for the Maximum Pairwise Difference.

6. For **(P2)**, when one of the components of $\boldsymbol{w}$ is 0, we have

$$MPD\left([\boldsymbol{w}^{\mathrm{T}}, 0]^{\mathrm{T}}\right) = MPD(\boldsymbol{w}).$$

   Therefore, Babies does not hold for the Maximum Pairwise Difference.

| | **(D1)** Robin Hood | **(D2)** Scaling | **(D3)** Rising Tide | **(D4)** Cloning | **(P1)** Bill Gates | **(P2)** Babies |
|---|---|---|---|---|---|---|
| $\mathbf{I}_{p,q}(\boldsymbol{w})$ | ✔ | ✔ | ✔ | ✔ | ✔ | ✔ |
| $Gini(\boldsymbol{w})$ | ✔ | ✔ | ✔ | ✔ | ✔ | ✔ |
| $MPD(\boldsymbol{w})$ | | | | ✔ | ✔ | |

Table 2: A Comparison of PQ Index ($\mathbf{I}_{p,q}(\boldsymbol{w})$), Gini Index ($Gini(\boldsymbol{w})$), and the Maximum Pairwise Difference ($MPD(\boldsymbol{w})$), in terms of the six ideal properties for sparsity measures (Hurley & Rickard, 2009). All six properties hold for the PQ Index and the Gini Index, whereas the Maximum Pairwise Difference only satisfies properties **(D4)** and **(P1)**.

### B.2 Proofs

*Proof of Theorem 3.1.* When $w_k \neq 0$ and $w_j = 0$ ($j \neq k$),

$$\mathbf{I}_{p,q}(\boldsymbol{w}) = 1 - d^{\frac{1}{q}-\frac{1}{p}} \cdot \frac{\|w_k\|_p}{\|w_k\|_q}$$
$$= 1 - d^{\frac{1}{q}-\frac{1}{p}}.$$

Thus, we complete the proof. □

*Proof of Theorem 3.2.* We utilize the property of Höder's inequality to prove the theorem:

**Lemma B.1** (Hölder's inequality). *For $a_i, b_i \in \mathbb{R}$ $(i = 1, \ldots, d)$, and $r_1, r_2 > 1$ such that $1/r_1 + 1/r_2 = 1$, we have*

$$\sum_{i=1}^d |a_i b_i| \leq \left(\sum_{i=1}^d |a_i|^{r_1}\right)^{\frac{1}{r_1}} \left(\sum_{i=1}^d |b_i|^{r_2}\right)^{\frac{1}{r_2}}.$$

*The equality holds if and only if there exists $\lambda \in \mathbb{R}$, such that*

$$|a_i|^{r_1} = \lambda \cdot |b_i|^{r_2}, \ i = 1, \ldots, n.$$

Take $r_1 = q/(q-p)$, $r_2 = q/p$, $a_i = 1$, and $b_i = w_i^p$ $(i = 1, \ldots, d)$. By Hölder's inequality,

$$\sum_{i=1}^d w_i^p \leq d^{\frac{q-p}{q}} \left(\sum_{i=1}^d w_i^q\right)^{\frac{p}{q}}$$

$$\iff \|\boldsymbol{w}\|_p \leq d^{\frac{1}{p}-\frac{1}{q}} \left(\sum_{i=1}^d w_i^q\right)^{\frac{1}{q}} = d^{\frac{1}{p}-\frac{1}{q}} \|\boldsymbol{w}\|_q$$

$$\iff \mathbf{I}_{p,q}(\boldsymbol{w}) \geq 0,$$

where the equality holds only when

$$w_1 = w_2 = \cdots = w_d.$$

Thus, we finish the proof. $\qquad\square$

*Proof of Theorem 3.3.* Since
$$\mathbf{I}_{1,2}(\boldsymbol{w}) = \mathbf{I}_{1,2}(\boldsymbol{w}/\|\boldsymbol{w}\|_2),$$

it suffices to show

$$\|\boldsymbol{w} - d^{-\frac{1}{2}} \cdot \mathbf{1}_d\|_2 = \sqrt{2\mathbf{I}_{1,2}(\boldsymbol{w})} \tag{6}$$

holds for a $\boldsymbol{w}$ with $\|\boldsymbol{w}\|_2 = 1$.

Let
$$m \triangleq \sqrt{d}\big(1 - \mathbf{I}_{p,q}(w)\big).$$

Then,
$$\|\boldsymbol{w}\|_1 = w_1 + \cdots + w_d = m \tag{7}$$

The intersection between the unit hypersphere and the above hyperplane is a hypersphere in $\mathbb{R}^{d-1}$. Denote it by

$$\mathcal{C} \triangleq \{\boldsymbol{w} \mid \|\boldsymbol{w}\|_1 = m, \|\boldsymbol{w}\|_2 = 1\}.$$

The normal vector of equation 7 is $d^{-1/2} \cdot \mathbf{1}_d$. The intersection between the normal vector and the hyperplane is the foot of the normal, which satisfies:

1. Its coordinates have the same value.

2. Its $l_1$ norm equals $m$.

Therefore, the foot of normal is $(m/d)\mathbf{1}_d$. Next, we will show that the foot of the normal is the center of $\mathcal{C}$. The proof will then be completed using the Pythagorean theorem, based on the distance between $\boldsymbol{w}$ and the foot of normal, as well as the distance between the foot of normal and $d^{-1/2} \cdot \mathbf{1}_d$.

We have for each point $\boldsymbol{w} \in \mathcal{C}$,

$$\left\|\boldsymbol{w} - \frac{m}{d}\mathbf{1}_d\right\|_2^2 = \|\boldsymbol{w}\|_2^2 + \left\|\frac{m}{d}\mathbf{1}_d\right\|_2^2 - \frac{2m}{d}\|\boldsymbol{w}\|_1$$

$$= 1 + \frac{m^2}{d} - \frac{2m^2}{d}$$

$$= 1 - \frac{m^2}{d},$$

namely, the distance from $\boldsymbol{w}$ to the foot of normal is $\sqrt{1 - m^2/d}$. Since the distance for all $\boldsymbol{w} \in \mathcal{C}$ to the foot of normal are the same, this point is the center of the hypersphere. The distance between the foot of normal and $d^{-1/2}\mathbf{1}_d$ is

$$\left\| \frac{m}{d}\mathbf{1}_d - \frac{1}{\sqrt{d}}\mathbf{1}_d \right\|_2 = \left\| \frac{m - \sqrt{d}}{d}\mathbf{1}_d \right\|_2 = \frac{\sqrt{d} - m}{\sqrt{d}},$$

where $m < \sqrt{d}$ since $m = \sqrt{d}\big(1 - \mathbf{I}_{p,q}(w)\big)$. Therefore, by Pythagorean theorem, the distance from each $\boldsymbol{w} \in \mathcal{C}$ to $d^{-1/2}\mathbf{1}_d$ is

$$\sqrt{\left( \frac{\sqrt{d} - m}{\sqrt{d}} \right)^2 + \left( \sqrt{1 - \frac{m^2}{d}} \right)^2} = \sqrt{2\big(1 - \frac{m}{\sqrt{d}}\big)} = \sqrt{2\mathbf{I}_{1,2}(\boldsymbol{w})}.$$

Therefore, we obtain Equation equation 6 and complete the proof. $\qquad\square$

*Proof of Theorem 3.4.* We prove this theorem by contradiction. First, we will show that if $\mathbf{I}_{1,2}(\tilde{\boldsymbol{w}}) = \mathbf{I}_{1,2}(\boldsymbol{w})$,

$$\tilde{c} = \left( w_1 - \frac{1}{d-1}\sum_{i=2}^{d} w_i \right) \cdot \frac{2(d-1)^2}{d^2 - d + 1}. \tag{8}$$

Then, we will prove that the above contradicts with the assumption that $\tilde{w}_1 = \tilde{w}_{max}$.

We have

$$(w_1 - \tilde{c})^2 + \sum_{i=2}^{d} \left( w_i + \frac{\tilde{c}}{d-1} \right)^2 = \|\boldsymbol{w}\|_2^2$$

$$\Longleftrightarrow w_1^2 - 2\tilde{c}w_1 + \tilde{c}^2 + \sum_{i=2}^{d} \left( w_i^2 + \frac{2\tilde{c}}{d-1}w_i + \frac{\tilde{c}^2}{(d-1)^2} \right) = \|\boldsymbol{w}\|_2^2$$

$$\Longleftrightarrow \sum_{i=1}^{d} w_i^2 - 2\tilde{c}\left( w_1 - \frac{1}{d-1}\sum_{i=2}^{d} w_i \right) + \frac{d^2 - d + 1}{(d-1)^2}\tilde{c}^2 = \|\boldsymbol{w}\|_2^2$$

$$\Longleftrightarrow \frac{d^2 - d + 1}{(d-1)^2}\tilde{c}^2 - 2\tilde{c}\left( w_1 - \frac{1}{d-1}\sum_{i=2}^{d} w_i \right) = 0$$

Let

$$q \triangleq \frac{d^2 - d + 1}{(d-1)^2}.$$

By the quadratic formula, we have

$$\tilde{c} = \frac{2\left( w_1 - \frac{1}{d-1}\sum_{i=2}^{d} w_i \right) + \sqrt{4\left( w_1 - \frac{1}{d-1}\sum_{i=2}^{d} w_i \right)^2}}{2q}$$

$$= \frac{2\left( w_1 - \frac{1}{d-1}\sum_{i=2}^{d} w_i \right)}{q} \quad \text{(we require } \tilde{c} \neq 0\text{)}.$$

Therefore, we obtain Equation equation 8.

Recall that after the transformation, $w_1$ will decrease by $\tilde{c}$ and the average of $w_2, \ldots, w_d$ will increase by $\tilde{c}/(d-1)$. Since

$$
\begin{aligned}
\tilde{c} + \frac{\tilde{c}}{d-1} &= 2\left(w_1 - \frac{1}{d-1}\sum_{i=2}^{d} w_i\right) \cdot \left(\frac{(d-1)^2}{d^2-d+1} + \frac{d-1}{d^2-d+1}\right) \\
&= 2\left(w_1 - \frac{1}{d-1}\sum_{i=2}^{d} w_i\right) \cdot \left(\frac{d^2-d}{d^2-d+1}\right) \\
&> w_1 - \frac{1}{d-1}\sum_{i=2}^{d} w_i \quad \text{(we assume } d \geq 2\text{)},
\end{aligned}
$$

$\tilde{w}_1$ is smaller than the average of $\tilde{w}_2, \ldots, \tilde{w}_d$, which gives a contradiction. Thus, we finish the proof. $\qquad \square$

*Proof of Theorem 3.5.* We first prove Inequality equation 1, then Inequality equation 2, finally Inequality equation 3.

**Proof of Inequality equation 1** We have

$$
\begin{aligned}
Gini(\boldsymbol{w}) &= \frac{\sum_{i=1}^{d}\sum_{j=1}^{d}|w_i - w_j|}{2d\sum_{i=1}^{d} w_i} \\
&\leq \frac{\sum_{i=1}^{d}\sum_{j=1}^{d} MPD(\boldsymbol{w})}{2d\sum_{i=1}^{d} w_i} \\
&= \frac{d^2 MPD(\boldsymbol{w})}{2d\|\boldsymbol{w}\|_1} \\
&= \frac{d MPD(\boldsymbol{w})}{2\|\boldsymbol{w}\|_1}.
\end{aligned}
$$

Since $\|\boldsymbol{w}\|_2 \leq \|\boldsymbol{w}\|_1$, we obtain Inequality equation 1.

**Proof of Inequality equation 2** By Theorem 3.3,

$$
\left|w_{max}/\|\boldsymbol{w}\|_2 - d^{-\frac{1}{2}}\right| \leq \sqrt{2\mathbf{I}_{p,q}(\boldsymbol{w})},
$$
$$
\left|w_{min}/\|\boldsymbol{w}\|_2 - d^{-\frac{1}{2}}\right| \leq \sqrt{2\mathbf{I}_{p,q}(\boldsymbol{w})}.
$$

By triangular inequality,

$$
\begin{aligned}
\|\boldsymbol{w}\|_2^{-1} MPD(\boldsymbol{w}) &= |w_{max}/\|\boldsymbol{w}\|_2 - w_{min}/\|\boldsymbol{w}\|_2| \\
&\leq \left|w_{max}/\|\boldsymbol{w}\|_2 - d^{-\frac{1}{2}}\right| + \left|w_{min}/\|\boldsymbol{w}\|_2 - d^{-\frac{1}{2}}\right| \\
&\leq 2\sqrt{2\mathbf{I}_{p,q}(\boldsymbol{w})}.
\end{aligned}
$$

Therefore, we obtain Inequality equation 2.

**Proof of Inequality equation 3** Since both PQ Index and Gini Index are scale-invariant, it suffices to show that the inequality holds for $\boldsymbol{w}$ satisfying $\|\boldsymbol{w}\|_1 = 1$. We will first prove that $\mathbf{I}_{1,2}(\boldsymbol{w})$ is convex. Next, we will show that each $\boldsymbol{w}$ can be expressed as a convex combination of $\ddot{\boldsymbol{w}}_1, \ldots, \ddot{\boldsymbol{w}}_d \in \mathbb{R}^d$, where

$$
\begin{aligned}
\ddot{\boldsymbol{w}}_1 &\triangleq [1, 0, 0 \ldots, 0]^{\mathsf{T}}, \\
\ddot{\boldsymbol{w}}_2 &\triangleq [1/2, 1/2, 0 \ldots, 0]^{\mathsf{T}}, \\
\ddot{\boldsymbol{w}}_3 &\triangleq [1/3, 1/3, 1/3 \ldots, 0]^{\mathsf{T}}, \\
&\vdots \\
\ddot{\boldsymbol{w}}_d &\triangleq [1/d, 1/d, 1/d \ldots, 1/d]^{\mathsf{T}},
\end{aligned}
$$

and that

$$\mathbf{I}_{1,2}(\ddot{\boldsymbol{w}}_j) \leq Gini(\ddot{\boldsymbol{w}}_j), \ \forall j \in \{1, \ldots, d\}.$$

*1) Convexity of $\mathbf{I}_{1,2}(\boldsymbol{w})$:*

We define

$$f(x) \triangleq 1 - d^{-1/2}\frac{1}{\sqrt{x}}, \quad x > 0,$$

which is monotonic increasing, and

$$g(\boldsymbol{w}) \triangleq \|\boldsymbol{w}\|_2.$$

Because the Hessian matrix of $g(\boldsymbol{w})$ is $\boldsymbol{I}$, $g(\cdot)$ is a convex function. Since

$$\mathbf{I}_{1,2}(\boldsymbol{w}) = f \circ g(\boldsymbol{w}),$$

which is the composition of a monotonic increasing function and a convex function, it is also convex.

*2) Properties of $\ddot{\boldsymbol{w}}_1, \ldots, \ddot{\boldsymbol{w}}_d$:* Without the loss of generality, we assume $w_1 \geq w_2 \geq \cdots \geq w_d$. Each $\boldsymbol{w}$ can be expressed as

$$\boldsymbol{w} = \alpha_1 \ddot{\boldsymbol{w}}_1 + \cdots + \alpha_d \ddot{\boldsymbol{w}}_d,$$

where

$$\alpha_d = dw_d,$$
$$\alpha_{d-1} = (d-1)(w_{d-1} - w_d),$$
$$\alpha_{d-2} = (d-2)(w_{d-2} - w_{d-1}),$$
$$\vdots$$
$$\alpha_1 = w_1 - w_2.$$

Since $w_1 \geq w_2 \geq \cdots \geq w_d$, the coefficients $\alpha_1, \ldots, \alpha_d \geq 0$. Also, by our assumption,

$$\|\boldsymbol{w}\|_1 = \|\ddot{\boldsymbol{w}}_1\|_1 = \cdots = \|\ddot{\boldsymbol{w}}_d\|_1 = 1,$$

we have $\sum_{i=1}^{n} \alpha_i = 1$. Therefore, $\boldsymbol{w}$ is a convex combination of $\ddot{\boldsymbol{w}}_1, \ldots, \ddot{\boldsymbol{w}}_d$. By convexity of $PQI(\boldsymbol{w})$, we have

$$\mathbf{I}_{1,2}(\boldsymbol{w}) \leq \alpha_1 \mathbf{I}_{1,2}(\ddot{\boldsymbol{w}}_1) + \cdots + \alpha_d \mathbf{I}_{1,2}(\ddot{\boldsymbol{w}}_d). \tag{9}$$

since

$$Gini(\boldsymbol{w}) = \frac{1}{d}\sum_{i=1}^{d}(d+1-2i)w_i$$

is a linear function of $\boldsymbol{w}$, we have

$$Gini(\boldsymbol{w}) = \alpha_1 Gini(\ddot{\boldsymbol{w}}_1) + \cdots + \alpha_d Gini(\ddot{\boldsymbol{w}}_d). \tag{10}$$

According to Inequality equation 9 and Equation equation 10, it remains to show for each $j \in \{1, \ldots, d\}$,

$$PQI(\ddot{\boldsymbol{w}}_j) \leq Gini(\ddot{\boldsymbol{w}}_j).$$

For each $j \in \{1, \ldots, d\}$, we have

$$PQI(\ddot{\boldsymbol{w}}_j) = 1 - \frac{1}{\sqrt{d}} \cdot \frac{1}{\sqrt{j \cdot j^{-2}}} = 1 - \sqrt{\frac{j}{d}},$$

and

$$Gini(\ddot{\boldsymbol{w}}_i) = \frac{d}{d} \cdot \sum_{i=1}^{d} w_i - \frac{1}{d} \cdot \sum_{i=1}^{j}(2i-1)j^{-1} = 1 - \frac{1}{d} \cdot \sum_{i=1}^{j}(2i-1)j^{-1} = 1 - \frac{j}{d}.$$

Since

$$\sqrt{\frac{j}{d}} \geq \frac{j}{d}, \ j \in \{1, \ldots, d\},$$

we complete the proof.

$\square$

*Proof of Theorem 3.6.* Since both PQ Index and Gini Index are scale-invariant, without loss of generality, we assume $\|\boldsymbol{w}^{(1)}\|_q = \|\boldsymbol{w}^{(2)}\|_q$. Then,

$$\mathbf{I}_{p,q}(\boldsymbol{w}^{(1)}) < \mathbf{I}_{p,q}(\boldsymbol{w}^{(2)})$$

$$\iff \|\boldsymbol{w}^{(1)}\|_p > \|\boldsymbol{w}^{(2)}\|_p$$

$$\iff \sum_{i=1}^{d} (w_i^{(1)})^p > \sum_{i=1}^{d} (w_i^{(2)})^p.$$

Since $w_{max}^{(1)} = w_{max}^{(2)}$ and $w_{min}^{(1)} = w_{min}^{(2)}$,

$$\|\dot{\boldsymbol{w}}^{(1)}\|_p > \|\dot{\boldsymbol{w}}^{(2)}\|_p.$$

Therefore, we obtain $\mathbf{I}_{p,q}(\dot{\boldsymbol{w}}^{(1)}) < \mathbf{I}_{p,q}(\dot{\boldsymbol{w}}^{(2)})$ and complete the proof. $\square$

### B.3 OVERVIEW OF FAIRNESS CRITERIA

| Problem | Criteria | Expression |
|---------|----------|------------|
| Classification | Statistical Parity | $\max_{y \in \mathcal{Y}} \max_{a,a' \in \mathcal{A}} \left\| \mathbb{E}(f(X_a) = y) - \mathbb{E}(f(X_{a'}) = y) \right\|$ |
| | $S$-Statistical Parity | $\max_{y \in \mathcal{Y}} S\big(\mathbb{E}(f(X_{a_i}) = y)_{i=1}^{|\mathcal{A}|}\big)$ |
| | Equalized Odds | $\max_{y,y' \in \mathcal{Y}} \max_{a,a' \in \mathcal{A}} \left\| \mathbb{P}_{y',a}(f(X) = y) - \mathbb{P}_{y',a'}(f(X) = y) \right\|$ |
| | $S$-Equalized Odds | $\max_{y \in \mathcal{Y}} S\big([g(f(X_{a_i}), Y_{a_i})]_{i=1}^{|\mathcal{A}|}\big)$ |
| Regression | Statistical Parity (*weak*) | $\max_{a,a' \in \mathcal{A}} \left\| \mathbb{E}(f(X_a)) - \mathbb{E}(f(X_{a'})) \right\|$ |
| | $S$-Statistical Parity (*weak*) | $S\big([\mathbb{E}(f(X_{a_i}))]_{i=1}^{|\mathcal{A}|}\big)$ |
| | Statistical Parity | $\sup_{y \in \mathcal{Y}} \max_{a,a' \in \mathcal{A}} \left\| \mathbb{P}_a(f(X) \leq y) - \mathbb{P}_{a'}(f(X) \leq y) \right\|$ |
| | $S$-Statistical Parity | $\sup_{y \in \mathcal{Y}} S\big([\mathbb{P}_{a_i}(f(X) \leq y)]_{i=1}^{|\mathcal{A}|}\big)$ |
| | Statistical Parity (*W*) | $\int_{y \in \mathcal{Y}} \max_{a,a' \in \mathcal{A}} \left\| \mathbb{P}_a(f(X) \leq y) - \mathbb{P}_{a'}(f(X) \leq y) \right\| dy$ |
| | $S$-Statistical Parity (*W*) | $\int_{y \in \mathcal{Y}} S\big([\mathbb{P}_{a_i}(f(X) \leq y)]_{i=1}^{|\mathcal{A}|}\big) dy$ |
| | Equalized Odds | $\max_{a,a' \in \mathcal{A}} \left\| g(f(X_a), Y_a) - g(f(X_{a'}), Y_{a'}) \right\|$ |
| | $S$-Equalized Odds | $S\big([g(f(X_{a_i}), Y_{a_i})]_{i=1}^{|\mathcal{A}|}\big)$ |

Table 3: Sparisty-based criteria ($S$-$*$) and their Maximum Pairwise Distance (MPD) counterparts.

We populate fairness criteria in Table 3 above for completeness. Formally, we define the following criteria under the Maximum Pairwise Difference and Sparsity.

**Definition B.1** (Statistical Parity (*weak*))**.** Given the regression output $f(X)$, the weak statistical parity is defined as

$$\max_{a,a' \in \mathcal{A}} \left\| \mathbb{E}(f(X_a)) - \mathbb{E}(f(X_a')) \right\|,$$

where $\mathbb{E}(\cdot)$ stands for the expectation over the input.

Next, we introduce three types of sparsity-based statistical parities for regression models: a weak statistical parity based on the model output $f(X)$, a statistical parity based on the cumulative distribution function (CDF) of $f(X)$, and a statistical parity that combines the CDF with integration.

**Definition B.2** ($S_r$-Statistical Parity (*weak*))**.** The weak sparsity-based statistical parity is measured by

$$S\big([\mathbb{E}(f(X_{a_i}))]_{i=1}^{|\mathcal{A}|}\big).$$

**Definition B.3** ($S_r$-Statistical Parity (*W*))**.** The integral sparsity-based statistical parity is defined as

$$\int_{y \in \mathcal{Y}} S\big([\mathbb{P}_{a_i}(f(X) \leq y)]_{i=1}^{|\mathcal{A}|}\big) dy. \tag{11}$$

This statistical parity measure is inspired by the Wasserstein distance (Kantorovich, 1960; Villani & Society, 2003), which quantifies the difference between two probability distributions by integrating the difference between their CDFs.

## C  EXPERIMENT IMPLEMENTATION

### C.1  DATASET DETAILS

**Classification.**  For classification task we benchmark on six datasets, with four binary classification datasets and two multi-classification datasets:

- *UCI Adult* (Murphy & Aha, 1996): The task in dataset is to use provided demographic features to predict whether someone's income is above 50k or not. Gender in the dataset is treated as the sensitive attribute. This dataset contains 48,842 instances. ($|\mathcal{Y}| = 2, |\mathcal{A}| = 2$)

- *COMPAS* (Angwin et al., 2016): The task involves predicting whether an individual is likely to reoffend based on their criminal history, time spent in prison, demographic information, and risk scores, with race (Caucasian vs. African-American) serving as the sensitive attribute. The dataset comprises 7,918 instances. ($|\mathcal{Y}| = 2, |\mathcal{A}| = 2$)

- *HSLS* (Ingels et al., 2011): High School Longitudinal Study contains 23,000+ education-related surveys collected from parents and students. It contains features such as demographic and school information of the students, as well as their academic performances from different school years. The binary target is whether a student's test score is among top 50% performer or not (Jeong et al., 2022), with a binary sensitive attribute race (Under represented minorty vs Asian/White). We use an preprocessed version encompassing 14,509 instances provided from Alghamdi et al. (2022) which filtered out entries with missing values from original data. ($|\mathcal{Y}| = 2, |\mathcal{A}| = 2$)

- *Enem* (INEP, 2020): This publicly available dataset is collected from 2020 Brazilian high school national exam and is consist of student demographics, socioeconomic questionnaire answers and exam scores. We preprocessed and randomly sampled 50k instances from the original dataset which contains around 5.8 million records, following the procedure in Alghamdi et al. (2022). Race is treated as the sensitive feature and binned grade as the target. ($|\mathcal{Y}| = 5, |\mathcal{A}| = 5$)

- *ACSIncome* (Ding et al., 2021): While UCI Adults datasets has been the major source for fairness research, this data is a superset of Adults derived from US Census survey and encompasses 1,664,500 entries. We construct this data with 5 races categories and 5 income categories. ($|\mathcal{Y}| = 5, |\mathcal{A}| = 5$)

- *Simulation*: Besides real world data, we include a simulated binary classification dataset with two groups and 10 features, each group contains 2,500 examples and examples in each group is drawn from Bernoulli distributions with $p = 0.5$ and $p = 0.8$. ($|\mathcal{Y}| = 2, |\mathcal{A}| = 2$)

- *Simulation (Multigroup)*: We simulated a binary classification dataset with an arbitrary number of sensitive groups to examine different criteria under intersectional fairness. Specifically, we aim to highlight the differences between metrics based on Maximum Pairwise Distance (MPD) and those based on sparsity. The total number of samples was fixed at 100,000, and we varied the number of groups ($n_g$) to generate corresponding datasets. For each group, the binary class weights were set to $0.5 - 0.4 \times \frac{g_i}{n_g - 1}$ and $0.5 + 0.4 \times \frac{g_i}{n_g - 1}$, where $g_i$ is the group index. This setup ensures that the maximum class weight difference across all groups remains 0.8 for both classes, with varying intermediate values depending on group size.

- *Adult (Multigroup)*: To evaluate intersectional fairness on the *Adult* dataset, we consider *gender* ($|A| = 2$), *race* ($|A| = 2$), and *age* (continuous) as candidate sensitive attributes. For the continuous variable *age*, we discretize observations into quantile-based bins to ensure approximately equal sample sizes across intervals. We experiment with age bins of sizes 2, 3, 4, and 5. To increase the number of sensitive groups, we construct attribute combinations: *gender*, *gender* × *race*, and *gender* × *race* × *age*, yielding dataset variants with 2, 10, 20, 30, 40, and 50 sensitive groups.

**Regression.** As for regression, we benchmark on two commonly used fair regression dataset, plus one simulated dataset:

- *Communities & Crime* (Redmond & Baveja, 2002): This dataset is about socioeconomics, crime data of US communities. The task is to predict number of violent crimes per 100,000 population using the provided features. We use race as the binary sensitive attribute (White vs non-White). The dataset contains in total 1,994 instances. ($|\mathcal{A}| = 2$)

- *LawSchool* (Ramsey & Wightman, 1998): This dataset is from Law School Admissions Councils National Longitudinal Bar Passage Study. The original datasets contains 22,407 records. After filtering out records with missing value and unknown races, it ends up having 20,053 instances. We make race as the sensitive attribute and predict the student's undergraduate GPA. ($|\mathcal{A}| = 4$)

- *Simulation*: Like classification, we include a simulated regression dataset with 1 feature. For each group, the feature is drawn from different gaussian distributions ( $\mathcal{N}(30, 4)$ vs $\mathcal{N}(10, 4)$) and target $Y$ is produced using the same coefficient but different $Var(\epsilon)$ (10 vs 1). ($|\mathcal{A}| = 2$)

## C.2 EXPERIMENTS DETAILS

For all datasets, we use an 80/20 split for training and testing, and conduct 10 independent experiments with different random seeds to evaluate performance. During model training, we apply feature normalization to improve training stability. Additionally, for regression tasks, a min-max transformation is applied to the target variable to standardize its range.

We used existing implementations for different bias-mitigation algorithms as they are available. Specifically, we use implementations from *AIF 360* library[1] for *Reduction*, *Rejection*, *EqOdds* and *CalEqOdds* in classification and *FairReg* in regression. For the other benchmark algorithms, we adapt implementations from their public code repositories.[2,3,4,5]

Below, we provide the hyperparameter range selected for each method that produces a trade-off curve in our benchmark. For those methods, the hyperparameter controls the tolerance of a fairness criteria violation.

| Method | Hyperparameter |
|---|---|
| *Reduction* (SP) | $0.0001, 0.01, 0.05, 0.5, 1, 3, 5, 10, 50, 100, 500$ |
| *Reduction* (EO) | $0.0001, 0.01, 0.05, 0.5, 1, 3, 5, 10, 50, 100, 500$ |
| *Rejection* (SP) | $0.001, 0.005, 0.01, 0.02, 0.05, 0.1, 0.2, 0.5, 1, 2$ |
| *Rejection* (EO) | $0.001, 0.005, 0.01, 0.02, 0.05, 0.1, 0.2, 0.5, 1, 2$ |
| *FairProj$_{KL}$* (SP) | $0.00, 0.001, 0.005, 0.01, 0.05, 0.1, 0.3, 0.5, 1.0$ |
| *FairProj$_{KL}$* (EO) | $0.00, 0.001, 0.005, 0.01, 0.05, 0.1, 0.3, 0.5, 1.0$ |
| *FairProj$_{CE}$* (SP) | $0.00, 0.001, 0.005, 0.01, 0.05, 0.1, 0.3, 0.5, 1.0$ |
| *FairProj$_{CE}$* (EO) | $0.00, 0.001, 0.005, 0.01, 0.05, 0.1, 0.3, 0.5, 1.0$ |
| *FairRR* (SP) | $0.00, 0.04, 0.08, 0.12, 0.16, 0.20, 0.40$ |
| *FairRR* (EO) | $0.00, 0.04, 0.08, 0.12, 0.16, 0.20, 0.40$ |
| *LinearPost$_C$* (SP) | $0.001, 0.01, 0.02, 0.04, 0.06, 0.08, 0.1, 0.12, 0.14, 0.16, 1000$ |
| *LinearPost$_C$* (EO) | $0.001, 0.01, 0.02, 0.04, 0.06, 0.08, 0.1, 1000$ |
| *LinearPost$_R$* | $0.00, 0.005, 0.01, 0.02, 0.05, 0.10, 0.15, 0.25, 0.30, 0.35, 0.45, 0.5, 1.0$ |
| *FairReg* | $0.001, 0.005, 0.01, 0.02, 0.05, 0.1, 0.4, 0.8, 0.99$ |

---

[1]*AIF360*: https://github.com/Trusted-AI/AIF360

[2]*FairProj*: https://github.com/HsiangHsu/Fair-Projection

[3]*LinearPost$_R$*: https://github.com/rxian/fair-regression

[4]*WassBC*: https://github.com/rxian/fair-regression (The *LineaPost* code repo also provides the python implementation for it)

[5]*LinearPost$_C$*: https://github.com/uiuctml/fair-classification

# D    ADDITIONAL RESULTS

## D.1    ABLATION STUDIES

In this section we conduct ablation studies for the proposed $S$-$*$ metric from the following perspectives: 1) Ensuring input values are positive, 2) Different $p$ and $q$ in PQI, 3) Different performance metric $g(\cdot)$, 4) Different $S(\cdot)$, 5) Different multi-class aggregations.

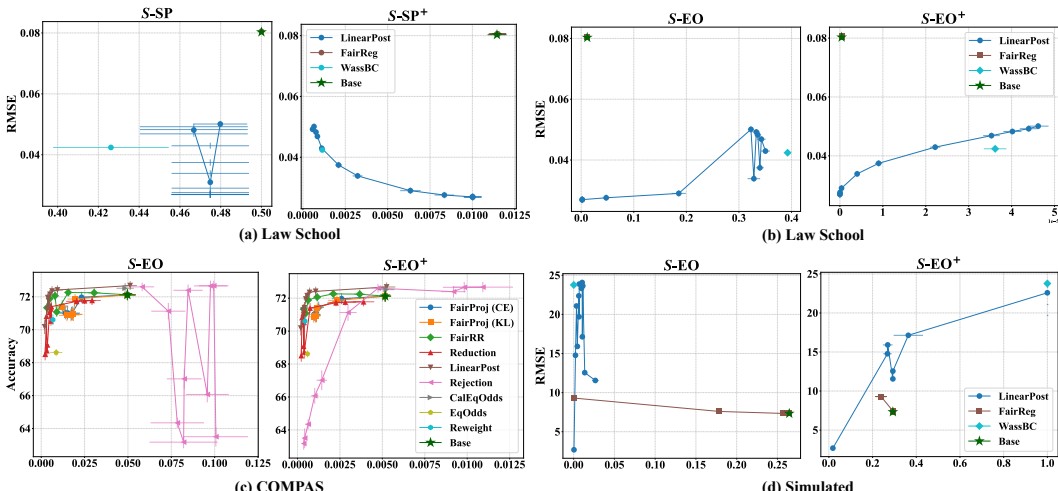

Figure 6: Results of applying $\boldsymbol{w} = \exp(\boldsymbol{w})$ to ensure positivity in different dataset and metrics.

**Positivity.**    Sparsity measure like the PQ Index is sensitive to the existence of 0 or extremely small values in the input, as they are measuring the ratio deviation. In addition, $S(\cdot)$ does not support negative values. In this ablation, we demonstrate the results when we ensure the positivity of the input values by applying the transformation $\boldsymbol{w} = \exp(\boldsymbol{w})$.

As shown in the Figure 6, the curves *LinearPost* and *Rejection* exhibit inconsistencies and high variability in the sparsity measure across multiple experiments due to extremely small values in confusion matrix (*Compas*), RMSE (*LawSchool*) or MSE loss (*LawSchool*) within a group. We show that by applying an exponential function to ensure all values are moderately positive, the expected trade-off curves can be recovered. Note that while this is a feasible solution in practice, other possible transformations still need further study.

**Performance Metric.**    We perform ablation studies on $g(\cdot)$ in $S$-EO by replacing it with other classification metrics computed from the F1 score, Area Under the Receiver Operating Characteristic Curve (AUROC) or a cross entropy loss function. We assess whether existing bias mitigation methods still work under these settings.  From the *HSLS* experiment results (Figure 7 (a)), we observe the trade-off curves when $g(\cdot)$ is specified as accuracy, F1 score or cross entropy loss. However, since the base classifier is already considered *fair* when performance metric $g(\cdot)$ is AUROC as the PQI Index is small, such a trade-off is not observed.

We include additional results from the remaining binary classification datasets in Figure 7 (b)-(d). As we empirically observe from the figure, these bias mitigation trade-off curves are generally preserved if the base classifier is not considered adequately fair (e.g., $S$-EO $\leq 10^{-4}$). In summary, understanding the robustness of these bias mitigation methods across various performance metrics $g(\cdot)$ still requires further efforts.

Besides the ablation study of $g(\cdot)$ in classification tasks, we also ablate $g(\cdot)$ in regression by replacing MSE with Mean Absolute Error (MAE) and R-squared ($R^2$). As shown in Figure 8, the results across multiple regression datasets suggest that different $g(\cdot)$ functions may present similar trade-off curves for $S$-EO when $g(\cdot)$ is specified as MAE. However, for $R^2$, the similar pattern is only observed in *LinearPost* and *WassBC*.

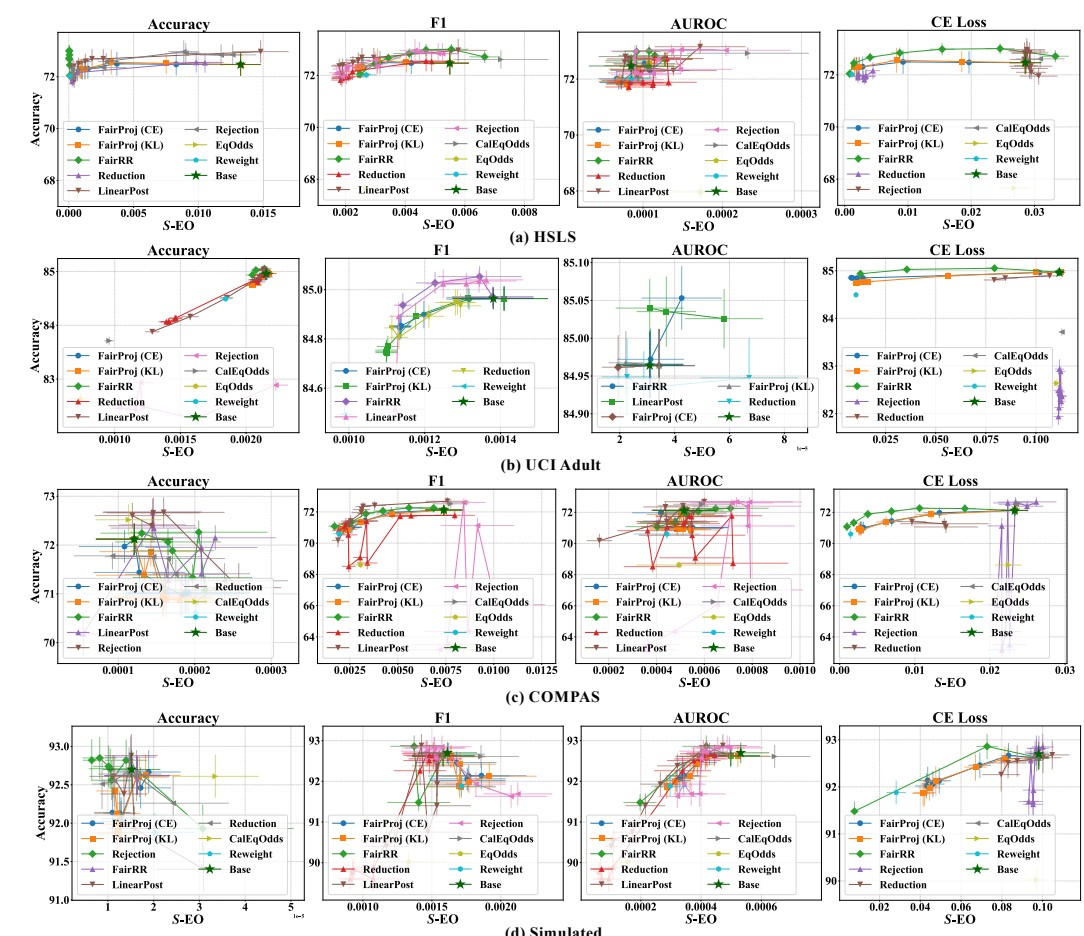

Figure 7: Replacing $g(\cdot)$ with other metrics in $S$-EO on various binary classification dataset. *LinerPost* does not provide outputs in probability space for CE loss calculation.

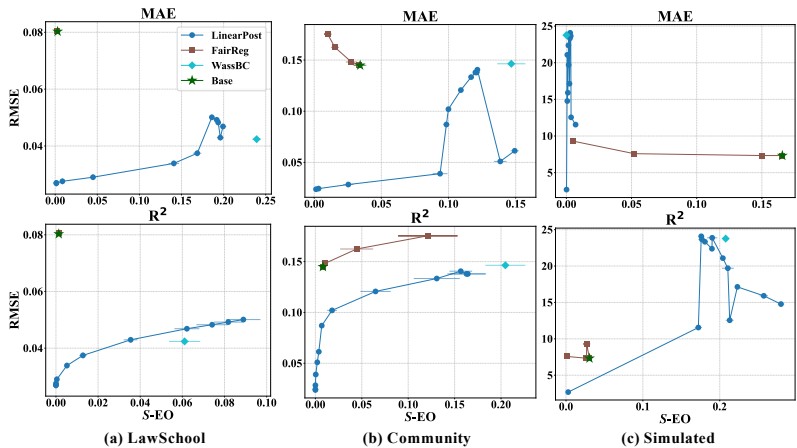

Figure 8: Replacing $g(\cdot)$ with other metrics in $S$-EO on regression datasets.

**PQ Ablation.** We incrementally change the values of $p$ and $q$ to check their effects on the resulting metrics. We include one example of ablating $p$ from $0.1$ to $0.9$ and $q$ from $1.1$ to $2.5$ in two classification datasets from applying *LinearPost* to examine the effects on the trade-offs. The results are shown in Figure 9. From the results we can see when $p$ and $q$ are closer to each other, it generates

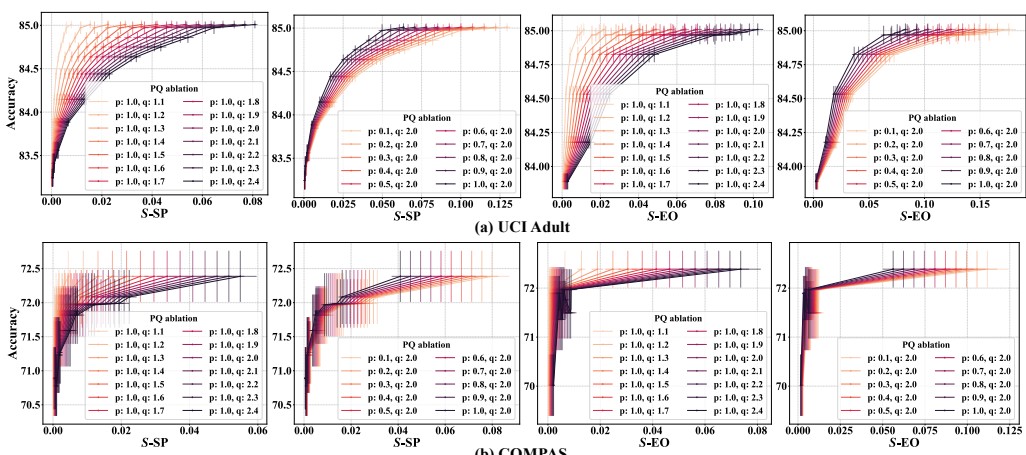

Figure 9: Ablation of $p$ and $q$ value on the output from *LinearPost* algorithm for the *UCI Adult* dataset ($|\mathcal{Y}| = 2, |\mathcal{A}| = 2$) and *COMPAS* dataset ($|\mathcal{Y}| = 2, |\mathcal{A}| = 2$).

values with a smaller scale. We also observe that the results have a smaller standard error across random data splits if $p$ and $q$ are closer.

**Gini Index.** In Figure 10, we demonstrate the effect of switching $S(\cdot)$ from PQ Index to Gini Index and show the effects on the simulated classification dataset. We observe almost identical pattern between the two sparsity measures, suggesting these two sparsity measures are intrinsically similar (Table 2).

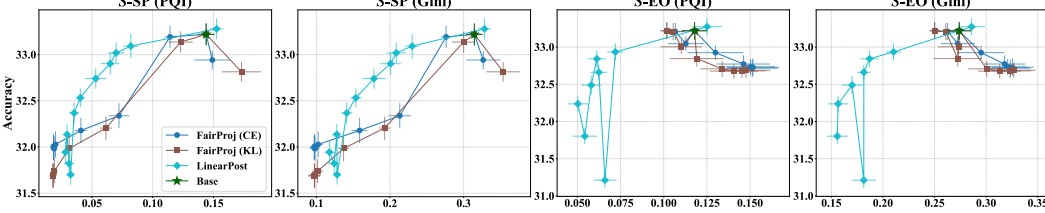

Figure 10: $S(\cdot)$ ablation comparisons on the *Enem* classification dataset ($|\mathcal{Y}| = 5, |\mathcal{A}| = 5$).

We present the trade-off curves on other datasets by switching PQ Index in $S$-$*$ to Gini Index and also observe the identical patterns between these two sparsity measures. Results are presented in Figure 11.

**Multi Class Aggregation.** In this ablation, we replace `max` with `mean` or `sum` for multi-class aggregations as previously mentioned (Section 4.1). In Figure 12 and Figure 13, we present results for one binary classification dataset, *UCI Adult* ($|\mathcal{Y}| = 2, |\mathcal{A}| = 2$), and one multi-class classification dataset, *Enem* ($|\mathcal{Y}| = 5, |\mathcal{A}| = 5$). The results suggest that, for all the classification fairness criteria we consider, their trade-off patterns remain invariant regardless of the multi-class aggregation operation used.

### D.2 ADDITIONAL EXPERIMENTAL RESULTS

We include additional results for the classification dataset (Figure 14) and regression dataset (Figure 15) in this section. Figure 14 demonstrates the comparison of different criteria on *COMPAS* ($|\mathcal{Y}| = 2, \mathcal{A} = 2|$), *Enem* ($|\mathcal{Y}| = 5, \mathcal{A} = 5|$), *HSLS* ($|\mathcal{Y}| = 2, \mathcal{A} = 2|$) and simulated classification

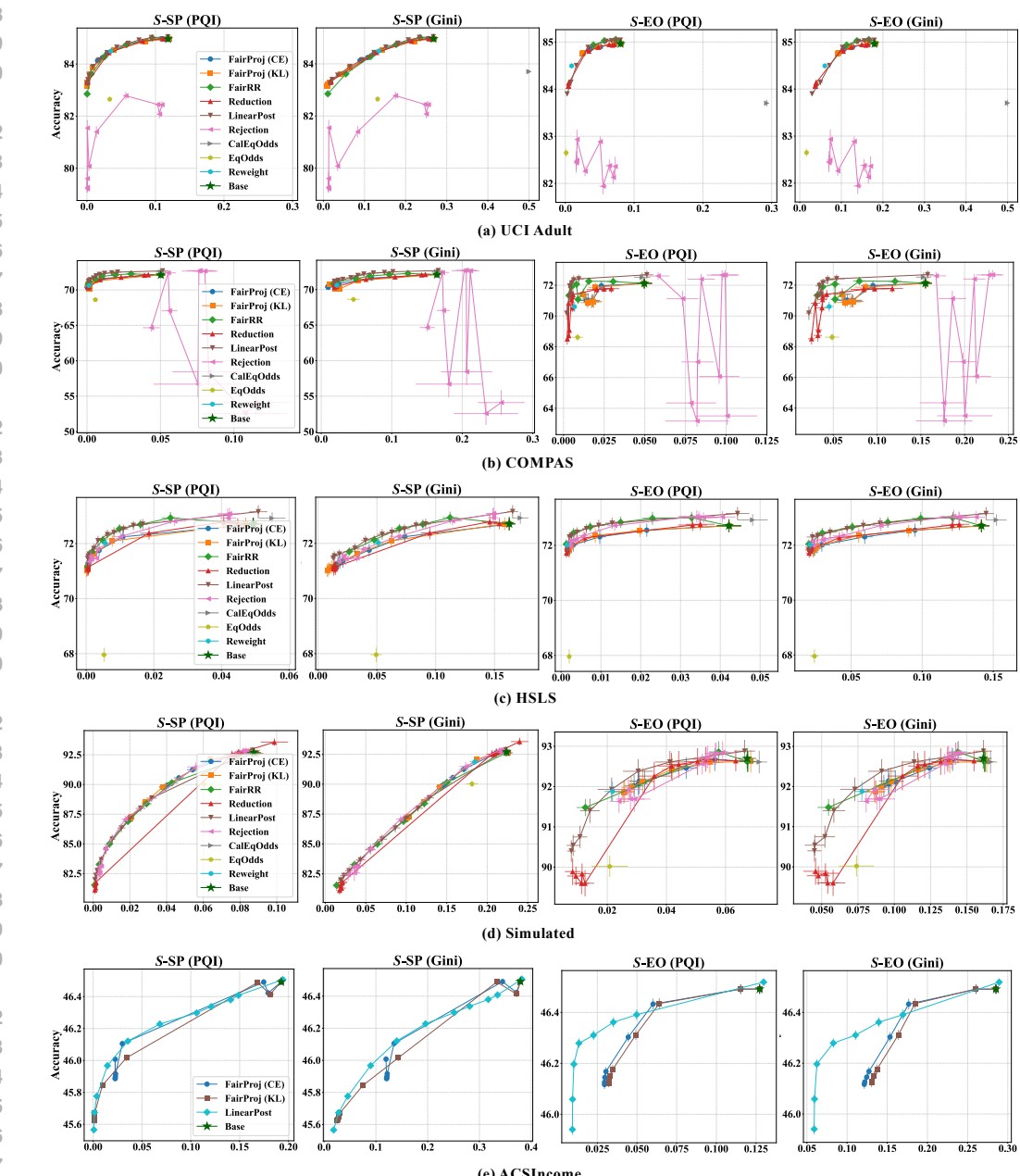

Figure 11: Comparison using PQ Index and Gini Index as $S(\cdot)$ in metric $S$-$*$.

$(|\mathcal{Y}| = 2, |\mathcal{A}| = 2|)$ datasets. The results of *LawSchool* $(|\mathcal{A}| = 4|)$ and the simulated regression dataset $(|\mathcal{A}| = 2|)$ are shown in Figure 15.

### D.3 SMOOTHNESS OF PQI

We show preliminary evidence in Figure 16 where we construct a simple constrained loss function to regularize on the model on the fairness: $\mathcal{L} = \mathcal{L}_{ce} + \lambda\mathcal{L}_g$, where $\mathcal{L}_{ce}$ is the cross entropy loss and $\mathcal{L}_g$ is the fairness regularization loss with weight $\lambda$. We conduct numerical optimization using an SGD optimizer on the Adult dataset. The results show that the PQ regularization loss converges more smoothly and can be optimized more effectively compared with the other two losses. This behavior is due to the smoother functional landscape of PQ, as illustrated in Figure 1.

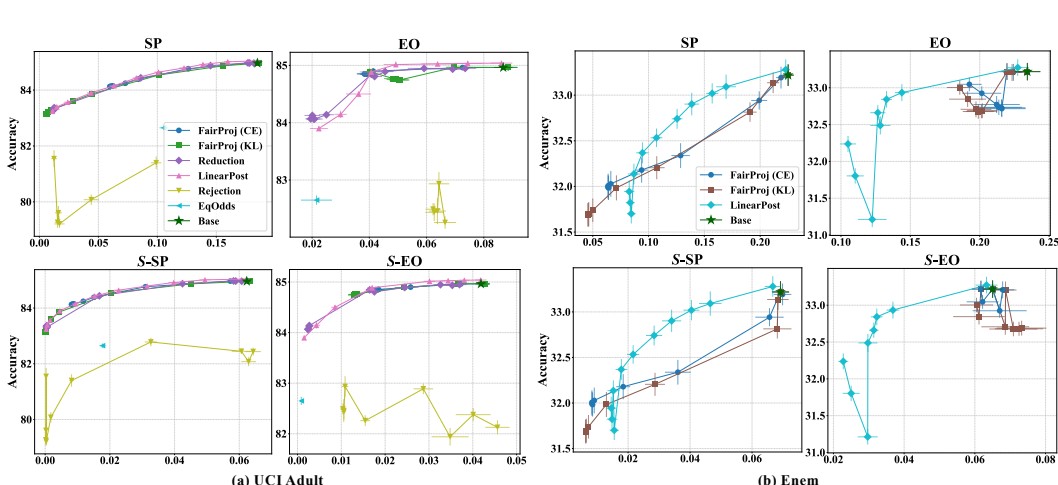

Figure 12: Replacing `max` with `mean` for multi-class aggregation.

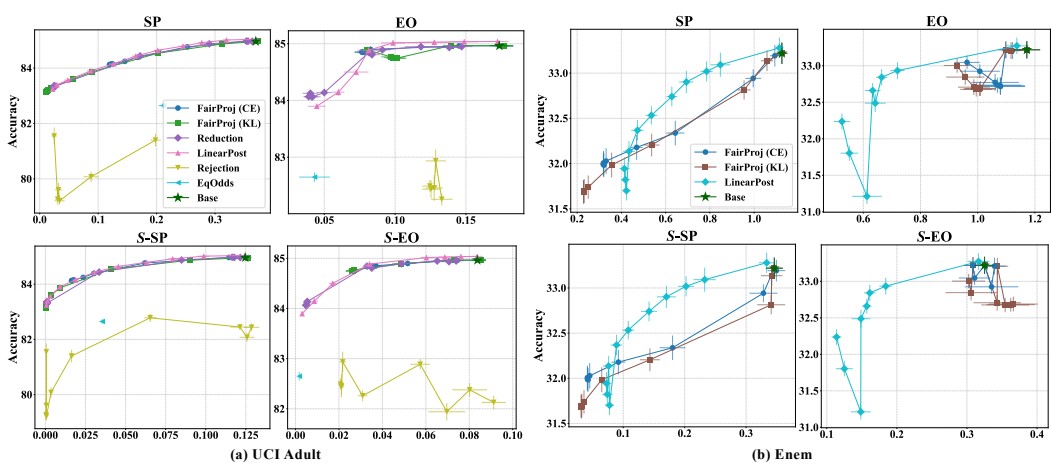

Figure 13: Replacing `max` with `sum` for multi-class aggregation.

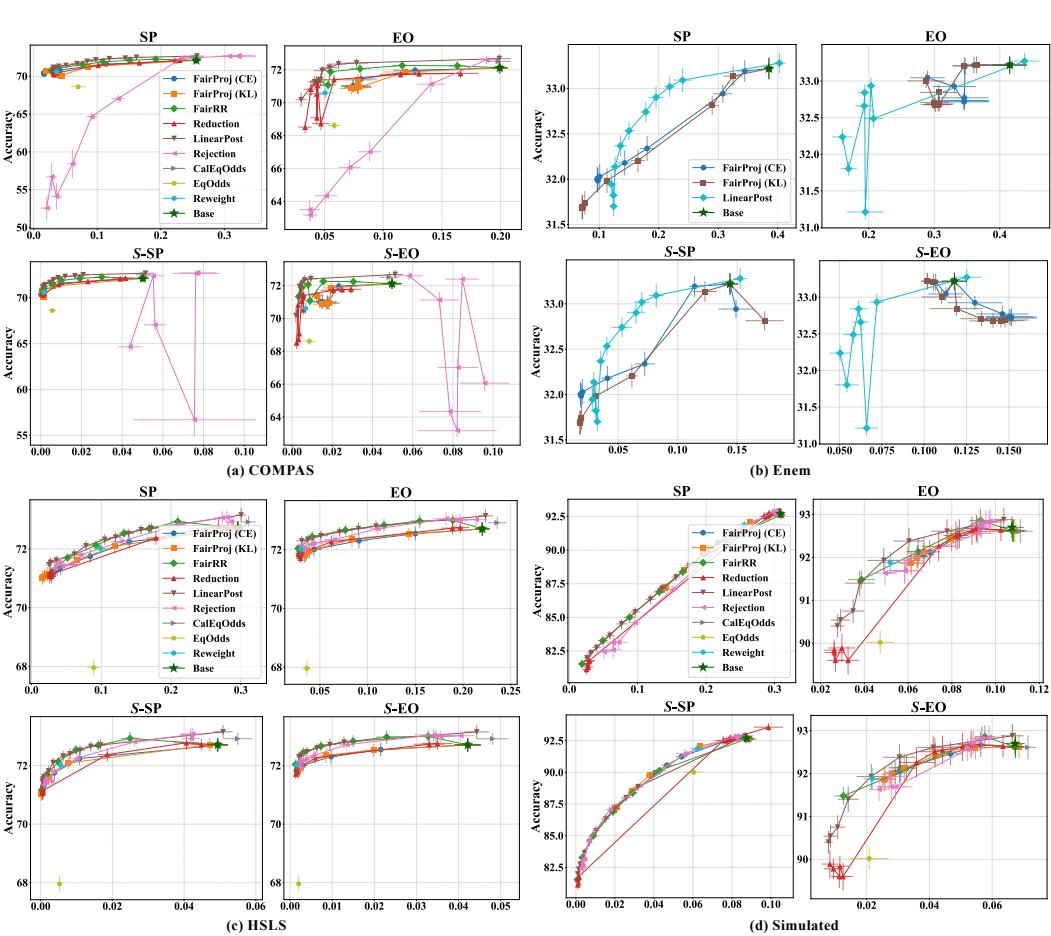

Figure 14: Comparison of sparsity criteria with baseline criteria in classification datasets.

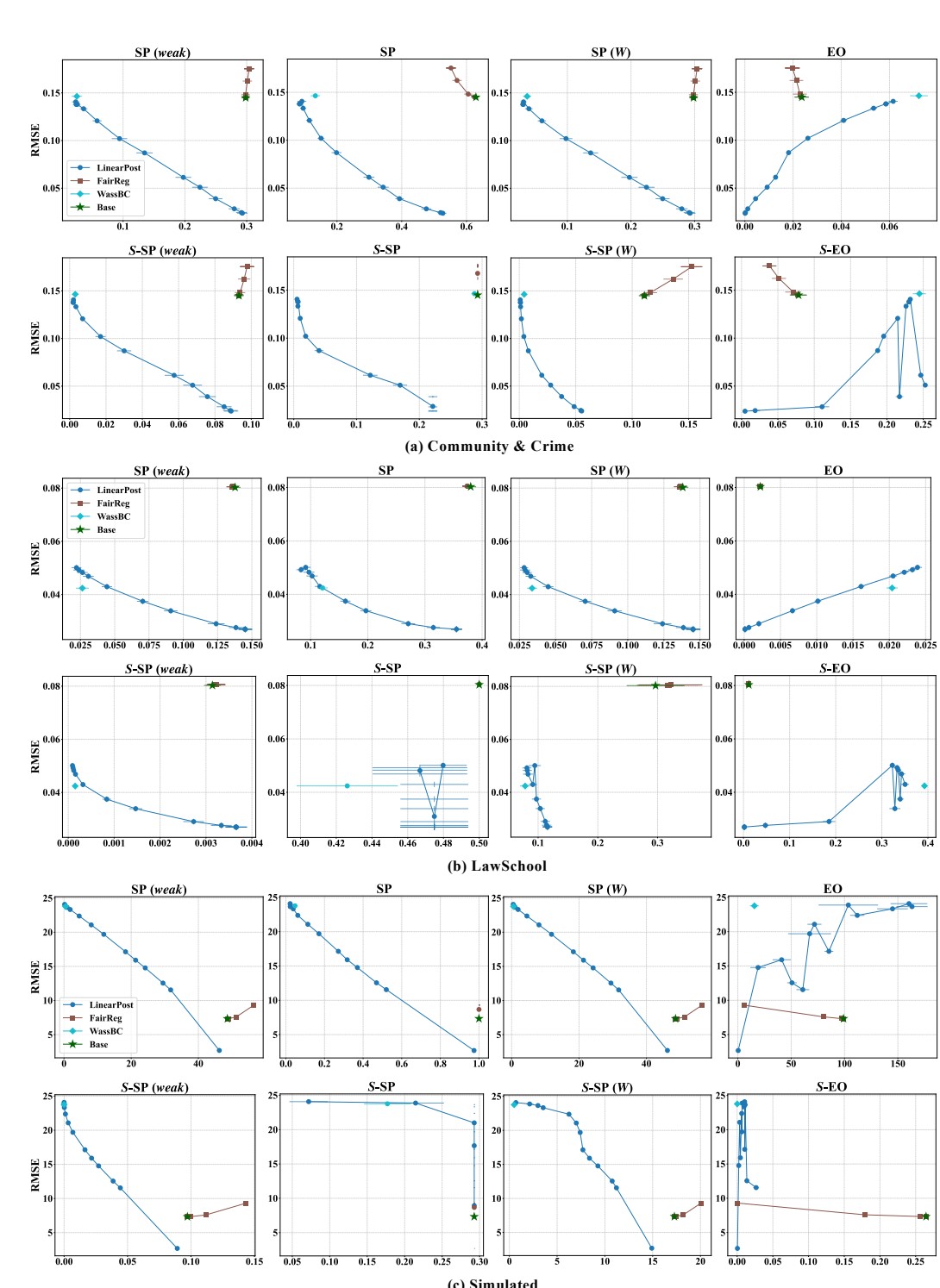

Figure 15: Comparison of all proposed sparsity criteria with baseline (MPD) criteria in three regression datasets.

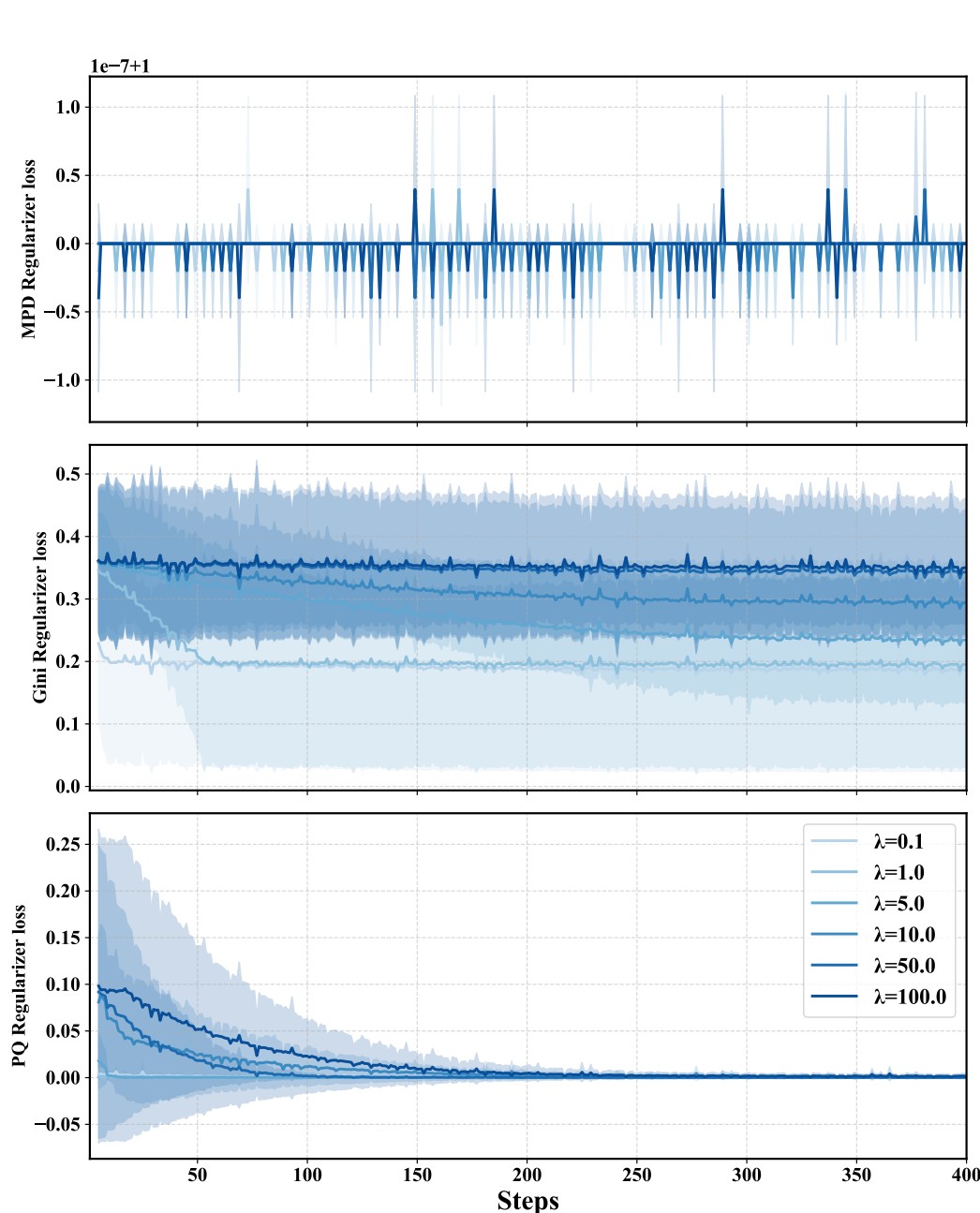

Figure 16: Regularizer loss trajectories for three types of regularization with SGD

