# OpenReview forum: "Toward Unifying Group Fairness Evaluation from a Sparsity Perspective"
_ICLR.cc/2026/Conference — Submitted to ICLR 2026_

### Official Review · Reviewer_7wmZ · 2025-10-21

**Soundness:** 2
**Presentation:** 2
**Contribution:** 2
**Rating:** 2
**Confidence:** 3

**Summary:**

The paper proposes a unified way to evaluate group fairness through sparsity. It studies links among Maximum Pairwise Difference, the Gini Index, and a PQ Index and argues that higher sparsity means lower fairness. Based on this view, it replaces the pairwise step in common criteria with a sparsity measure and defines S-SP and S-EO for classification and regression, with formulas and properties for PQ. Experiments across several datasets and bias mitigation methods show similar trends to MPD-style metrics and some differences in intersectional settings. The paper positions the work as an evaluation framework rather than a training algorithm.

**Strengths:**

1. Clear and unified formulation that maps standard group fairness metrics to a sparsity template and covers classification and regression including multi-class and multi-group cases. Table 1 makes the mapping explicit.

2. Useful theoretical connections between PQ, Gini, and MPD with properties and bounds that help interpret behavior of the proposed metrics.

3. Broad empirical sweep with classic classification datasets and two regression benchmarks, plus an intersectional study that highlights where sparsity criteria move differently from MPD.

**Weaknesses:**

1. The main claim is largely about reparameterizing MPD-style gaps into a norm-based sparsity score. The conceptual novelty relative to prior distributional fairness measures and to existing uses of Gini-type indices is not sharply distinguished. More head-to-head comparisons to recent distribution level metrics would help.

2. Practical stability relies on an exponential transform to handle negative or near-zero entries before applying sparsity, which introduces an extra design choice and can affect scale and sensitivity. The paper acknowledges this issue but does not fully analyze when the transform changes conclusions.

3. The framework depends on selecting the function g for EO-type measures and on the sparsity parameters p and q, yet there is limited guidance on how to choose them or how results vary. Clear sensitivity curves for p and q, and for alternative g choices, are missing.

4. The experiments use simple base models and many post-processing mitigations. Evidence that the conclusions hold for stronger modern models or end-to-end in-processing methods is limited. A study with neural encoders would raise confidence in generality.

5. External validity is not fully assessed. The work argues that sparsity better reflects full distribution equity, yet there is no application level validation, such as calibration checks, threshold sensitivity, or decision utility under policy constraints.

**Questions:**

1. How sensitive are S-SP and S-EO to the choice of p and q in the PQ Index. Please report curves for p and q on classification and regression datasets.

2. For S-EO you instantiate $g$ as a mix of TPR and FPR. What happens if $g$ is accuracy, cross entropy, or a proper scoring rule? Do rankings by the sparsity metric change?

3. How often does the exponential transform change the ordering of methods relative to raw MPD or to sparsity computed on untransformed nonnegative surrogates. Can you add a robustness study?

4. In the intersectional study, can you quantify when S-SP diverges from SP and link this to Theorem 3.6 with synthetic controls? A simple simulation that holds max and min fixed while varying the middle components would be helpful.

5. Could you evaluate the framework with stronger base learners and an in-processing algorithm to show that the alignment trends are not an artifact of linear models and post-processing?

---

> ### Author Response · Authors · 2025-11-28
>
> We thank reviewer for examining our paper and the feedback. We address each of the concern as below:
>
> > Response to Weakness 1: More head-to-head comparisons to recent distribution level metrics would help...
>
> Existing work on distributional fairness metrics is relatively limited, and prior approaches primarily focus on how internal distributional statistics are computed [1]. In contrast, our study centers on group-wise comparisons, which represent a fundamentally different perspective. To the best of our knowledge, no prior work has explicitly adopted this group-comparison viewpoint. We therefore welcome any relevant references the reviewers may be aware of.
>
> > Response to Weakness 2/Question 3: Practical stability relies on an exponential transform ...
>
> We respectfully disagree with this evaluation. Ratio-based metrics are inherently unstable when denominators approach zero. As demonstrated in the ablation study presented in Figure 5, the exponential transformation effectively restores the expected behavior of the sparsity-based metric under such conditions. While we acknowledge that many alternative transformations could be considered, a comprehensive investigation of all possible variants lies beyond the scope of this work.
>
> > Response to Weakness 3/Question 1/Question 2: The framework depends on selecting the function g for EO-type measures...
>
> Our framework provides an interface that allows users to specify the function $g$ according to the metric of interest, which is precisely why we decouple internal metric computations from group-wise comparisons. To illustrate how different choices of $g$ influence the behavior of the sparsity-based metric, we include ablation studies and interpret the corresponding results in Figure 6 (varying $g$ in classification) and Figure 7 (varying $g$ in regression). The general recommendation is to set $p = 1$ and $q = 2$, based on the theoretical properties established in Section 3.1. Empirically, Figure 8 (Appendix) further shows that as long as the condition $0 < p \leq 1 < q$ is satisfied, varying $p$ and $q$ influences only the scale of the metrics, while the overall alignment patterns remain unchanged.
>
> > Response to Weakness 4/Question 5: The experiments use simple base models...
>
> We respectfully disagree with this claim. Since fairness metrics depend solely on model outputs, their values are independent of the particular model architecture or implementation used to generate those outputs.
>
> > Response to Weakness 5: External validity is not fully assessed...
>
> It is unclear to us how the referenced terms (such as calibration checks, threshold sensitivity, and decision utility) are intended to apply within the fairness context of our work. These concepts carry different meanings across subfields, and without further clarification, it is difficult to ascertain the specific concerns being raised. We would greatly appreciate it if the reviewer could provide additional context or relevant references so that we may address the point appropriately.
>
> > Response to Question 4: In the intersectional study, can you quantify when S-SP diverges from SP and link this to Theorem 3.6 with synthetic controls?...
>
> We conduct this experiment in Section 5.3, where we show that S-SP and SP can diverge under two different conditions and We show that S-SP and SP can diverge under two different conditions and refer the reader to Theorem 3.6 (line 459) for the corresponding theoretical justification. Specifically, in our simulated setting, we fix the maximum and minimum class weights and interpolate the intermediate class weights, illustrating that S-SP varies depending on these middle components.
>
> ---
> [1] Retiring $\Delta$DP: New Distribution-Level Metrics for Demographic Parity

---

### Official Review · Reviewer_GwDM · 2025-10-24

**Soundness:** 3
**Presentation:** 3
**Contribution:** 3
**Rating:** 6
**Confidence:** 4

**Summary:**

The paper presents a novel framework for fairness evaluation based on sparsity. The authors first propose the use of the PQ index, originally introduced for pruning, as a sparsity measure for fairness evaluation, in a manner similar to the Gini Index. They then describe the properties of this index in comparison to the Gini Index, including differences with respect to the Maximum Pairwise Difference (MPD). The paper further outlines currently used fairness metrics based on MPD and suggests replacing MPD with alternative sparsity measures such as the Gini or PQ index.
The authors demonstrate that the behavior of the proposed metrics aligns with that of standard fairness metrics when applied to a binary sensitive attribute and bias mitigation algorithms. Moreover, they show that these sparsity-based metrics are better suited for capturing fairness in scenarios where the sensitive attribute consists of multiple groups. This is because both the Gini and PQ indices consider the full vector of group values, rather than just the maximum and minimum, and thus capture disparities more effectively.

**Strengths:**

The paper is well-motivated, addressing an important topic in fairness, the need for appropriate metrics that effectively capture the disparity we aim to measure. It is well-written and features extensive experiments that consider multiple datasets and bias mitigation techniques. The motivation for using sparsity-based metrics in the context of intersectional fairness is compelling and well-justified, and the results of that section (5.3) are convincing.

**Weaknesses:**

The actual contribution of the proposed metric should be better highlighted from the start. Specifically, from what I understand from the subsequent sections, the main advantages are that the metric can be consistently applied to both binary and multi-class settings, and that it captures disparities across the entire group distribution more effectively than MPD in multi-class scenarios. However, the abstract and introduction make broader claims such as “generalizability across different machine learning problems” and “applicability to a wide range of machine learning tasks.” While these claims may be valid, they should be better supported and clarified in light of the results presented. A dedicated discussion section elaborating on the observed advantages of the proposed framework compared to existing metrics would help strengthen this point.
Furthermore, from how the introduction is written now, I would understand that the framework is unified for both classification and regression tasks, but Section 4 introduces two distinct metrics for these problem types. This discrepancy should be addressed to clarify the extent of the framework’s unification.
Lastly, while the idea of repurposing the PQ index, originally designed for pruning, in the context of fairness evaluation is promising, it is not clearly established what advantages it offers over the Gini index, given that the two share many of the same properties. The authors should better describe whether and how, in the fairness context, this measure captures aspects that the Gini index fails to capture. Additionally, the properties of the PQ index described in Section 3 largely mirror those from the original paper and, from my understanding, are not all essential for understanding the framework’s advantages.
As such, this section could be condensed to improve focus and clarity.

**Questions:**

What are the advantages of using the PQ index instead of the Gini Index? Does it capture something that the Gini Index doesn’t capture? In what terms is the framework “unified” and how is it backed up by the experiments?

---

> ### Author Response · Authors · 2025-11-28
>
> We sincerely thank the reviewer for their insightful comments and detailed feedback.
>
> > Response to Weakness: The actual contribution of the proposed metric should be better highlighted ...
>
> **Clarify the Contribution:** We particularly appreciate the suggestion to better highlight the core contributions of our proposed framework. We revised the Abstract section with more precise languages to underline the key contribution of this paper and summarized the finding of the benefits of using the sparsity as an alternative for fairness evaluation. The revised part is colored blue in the paper. The main contributions of our work are as follows:
> - A theoretically grounded unification of group fairness evaluation that decouples internal metric computations from group-wise comparisons.
> - The introduction of sparsity-based group comparisons as an alternative approach for capturing the full dynamics of group-level disparities.
> - A comprehensive theoretical analysis of both the shared properties and key distinctions between sparsity-based metrics and MPD, supported by extensive experiments across diverse supervised learning tasks that validate these theoretical findings.
>
> **Clarify the Unified Framework:** In our paper, the unified framework is derived by decoupling the computation of fairness metrics. Following this decoupling, we express the general formulation as $S([m_i]_{i=1}^{|A|})$, where $S$ denotes a group-level operator and $[m_i]$ is a vector of metrics computed from model outputs.
>
> For both classification and regression tasks, the internal metrics can be defined directly from model outputs as $m_i = \mathbb{E}(f(X_i) = y \mid A)$, which corresponds to the notion of Statistical Parity. For regression tasks, we can additionally utilize the cumulative distribution function (CDF) as the internal metric, leveraging the use of the Kolmogorov-Smirnov distance to capture distributional differences. It is important to note that the CDF is not defined for categorical variables in classification settings.
>
> The outer function $S(\cdot)$ is specified as a sparsity measure. When MPD is chosen as $S(\cdot)$, the resulting formulation recovers the existing baseline fairness metrics. In the experimental section, we include benchmarks covering: 1) binary groups with binary classification, 2) multiple groups with multi-class classification, 3) binary groups with regression, and 4) multiple groups with regression, thereby demonstrating the broad applicability of the unified framework.
>
> **Originality of Theorems in Section 3:** We argue that our study offers substantive new insights and theoretical contributions that extend well beyond the findings of Diao et al. (2023) [1]. While the original work introduces the PQ Index primarily as a compressibility measure for network pruning, we are the first to demonstrate its potential relevance for group fairness evaluation. As noted in line 142, Theorems 3.1 through 3.4 formally establish the conditions under which the PQ Index can be interpreted as a fairness metric, and Theorems 3.5 and 3.6 further characterize its theoretical relationships with MPD and the Gini Index. With the exception of Theorem 3.1, all remaining results, including Theorems 3.2 through 3.6 and their accompanying analyses, are original and novel contributions of this work.
>
> **PQ Index vs Gini Index:** Numerically, the PQ Index and the Gini Index exhibit distinct functional landscapes, as shown in Figure 1 of the paper. Future bias mitigation methods may benefit from the smoother behavior of the PQ Index relative to the Gini Index. Detailed numerical comparisons are provided in Appendix D.3 of the revised manuscript, with corresponding updates highlighted in blue. When used as evaluation metrics, the PQ Index and the Gini Index yield consistent qualitative trends, as demonstrated by our side-by-side comparison in the Appendix (line 1373).
>
> We emphasize the Gini Index in our paper because it is a well-established measure of inequality in the social sciences and is grounded in ethical theory, particularly in relation to Rawls’ principle of Fair Equality of Opportunity. Furthermore, the Gini Index can also be interpreted as a sparsity measure, which further motivates our use of sparsity-based formulations for quantifying algorithmic fairness in machine learning.
>
> ---
> [1] “Pruning deep neural networks from a sparsity perspective”

---

### Official Review · Reviewer_Nm1h · 2025-10-30

**Soundness:** 2
**Presentation:** 2
**Contribution:** 1
**Rating:** 2
**Confidence:** 4

**Summary:**

This paper experimentally examines the use of the PQ-index [1] in place of max pairwise distances (MPD) in two fairness criteria (statistical parity and equalized odds).   The comparison is performed on 6 datasets used for fair classification and regression.  Experimental results show that the baseline and sparsity-based measures of fairness have similar tradeoff curves between model performance and fairness.  Experiments examining intersectional fairness were done on a single dataset.  Authors claim these results suggest that sparsity-based fairness metrics may be more sensitive to heterogeneity in the groups.

**Strengths:**

+ Since sparsity measures have been used in studies on social inequality/fairness, the use of these measures in studying algorithmic fairness seems to be warranted.
+ Interesting theoretical results characterizing the properties of sparsity measures.

**Weaknesses:**

-	Paper did not clearly state what benefits they found in their examination.  Experimental results on the 6 datasets seemed to indicate both baseline fairness metrics and sparsity-based measures had similar tradeoff curves regarding model performance and fairness.
-	While there were results on a single dataset suggesting that sparsity-based metrics were “better” at handling group heterogeneity (sec 5.3).  I felt the experiments were too limiting to draw strong conclusions for or against the use of sparsity-based methods.  Only a single dataset was used.  The assertions made in the text regarding the robustness of the sparsity-based metric were not clearly tied back to the experimental results.

**Questions:**

Questions

1)	What is the benefit of using a sparsity-fairness metric over the baseline metrics?

Additional Remarks:

1)	Paper would benefit from a clear declaration of the need, benefits and weaknesses of using sparsity-based fairness metrics.

2)	I found the experiments in section 5.3 to be too limited to be convincing.  Assertion made in the section 5.3 that baseline metrics exhibit “inconsistent debiasing performance” was not clearly tied back to Fig 4 results.  Perhaps this is true regarding the simulated binary classification data, but I don’t see how you could reach this conclusion on the Adult data in Fig. 4.  I did not understand how one arrived at the assertion at the end of section regarding the greater robustness of sparsity-based metrics.  This section would benefit from more extensive testing with additional datasets as well as clearer explanations of how the section’s conclusions were made.  As it currently reads, the assertions regarding the potential +’s of sparsity-based measure regarding intersectional fairness seem vague and not clearly substantiated by the experiment.

References:

[1] Diao et al., “Pruning deep neural networks from a sparsity perspective”, arXiv preprent 2301.05601, 2023

---

> ### Author Response · Authors · 2025-11-28
>
> Thanks for your time reviewing our paper and making valuable feedback. We address each of your concerns as the following:
>
> > Response to Weakness 1 / Question 1 / Remark 1: Benefits and weakness of the proposed metric...
>
> The benefits and limitations of the sparsity-based metric must be interpreted within context, as demonstrated in Section 5.2, Section 5.3, and the additional Discussion section in the revised PDF.
>
> **Clarifying contributions**: To clarify, the primary goal of this paper is **not** merely to demonstrate the advantages of sparsity-based metrics over baseline fairness metrics. The main contributions of our work are as follows:
> - A theoretically grounded unification of group fairness evaluation that decouples internal metric computations from group-wise comparisons.
> - The introduction of sparsity-based group comparisons as an alternative approach for capturing the full dynamics of group-level disparities.
> - A comprehensive theoretical analysis of both the shared properties and key distinctions between sparsity-based metrics and MPD, supported by extensive experiments across diverse supervised learning tasks that validate these theoretical findings.
>
> **Fairness metrics are subjective** Given that evaluation criteria in ML fairness are inherently **subjective** and often **case-dependent**, our work does not claim that the sparsity-based metric is universally superior to existing fairness metrics. We propose it as an alternative for settings that prioritize fairness across all groups, reflected by the Gini Index and the PQ Index, rather than worst-case outcomes, reflected by MPD. In the paper, we first demonstrate experimentally that the proposed sparsity metric aligns closely with established benchmarks. This observation is consistent with our theoretical analysis of the common properties shared by sparsity and MPD, supporting the validity of using sparsity to quantify group fairness. We then design and conduct experiments to illustrate scenarios in which these two metrics diverge in behavior (see Response to Weakness 2 for details).
>
>
> > Response to Weakness 2 / Remark 2: Insufficient experiments & Clarity of Figure 4...
>
> **Additional Experiments:** In addition to the two cases presented in Section 5.3, we conduct an additional case study on recommendation systems to examine scenarios in which the two metrics diverge in behavior. The results show that the MPD-based metric is not only highly sensitive to outliers introduced by new samples but can also become dominated by them, whereas the sparsity-based metric remains comparatively stable. Such stability is theoretically expected, given that new samples are drawn uniformly and the pretrained model remains unchanged. The full experimental setup and corresponding results are provided in the Fair Recommendation System section of the revised PDF, with relevant updates highlighted in blue.
>
> **Robustness of $S-SP$:** From the bottom row of Figure 4, we observe a sudden increase in the SP value and its standard deviation for $LinearPost_{0.001}$ at a group size of 50. This behavior is caused by extreme class imbalance in the model predictions. The table below provides an illustrative example:
>
> | **Group id** | **+** | **-** |
> |--------------|-------|-------|
> | group 1      | 5     | 3     |
> | group 2      | 4     | 4     |
> | ...          |       |       |
> | group 19     | 6     | 0     |
> | ...          |       |       |
>
> In this scenario, the statistical parity of group 19 is 1 by definition, and the MPD-based metric preserves this maximum disparity in its final evaluation. In contrast, the sparsity-based metric naturally mitigates the effect of such anomalies. This observation supports our conclusion in line 475 that the sparsity-based metric provides greater robustness under severe in-group class imbalance.

---

### Official Review · Reviewer_NYpV · 2025-11-01

**Soundness:** 3
**Presentation:** 3
**Contribution:** 2
**Rating:** 4
**Confidence:** 4

**Summary:**

This paper proposes a unified framework for evaluating algorithmic fairness through sparsity measures. The authors theoretically analyze the PQ Index as a sparsity measure, establish its relationships with MPD, and reformulate classical fairness metrics (SP and EO) in terms of sparsity. Experiments on multiple datasets and with several bias mitigation methods demonstrate empirical alignment between sparsity-based and traditional fairness measures.

**Strengths:**

1. Using sparsity to connect different group-fairness metrics is an interesting idea that is rarely explored in existing fairness literature.
2. The theoretical section is well-written and mathematically grounded.

**Weaknesses:**

1. Limited novelty beyond reinterpretation. While the sparsity–fairness connection is elegant, the framework largely rephrases existing fairness measures in new mathematical form. The advantage of the proposed framework is more evident in intersectional fairness settings. Nonetheless, beyond these intersectional cases, the contribution remains primarily interpretive rather than methodological. There is little evidence that the proposed sparsity-based metrics yield different conclusions compared to MPD-based ones, apart from minor smoothness or stability differences.
2. The analysis of PQ Index properties relies heavily on prior results (Diao et al., 2023). The new theorems mostly restate or slightly adapt known properties to fairness interpretation rather than offering fundamentally new mathematical results.
3. There should be a constraint (\forall) for $j$ in Theorem 3.1 to indicate all elements other than $w_k$ is 0.

**Questions:**

Is there a principled way to select the sparsity parameters (p, q) for PQ Index beyond using (1, 2)?

---

> ### Author Response · Authors · 2025-11-28
>
> We sincerely appreciate your constructive feedback. We address your specific concerns below:
>
> > Response to Weakness 1: Limited novelty beyond reinterpretation...
>
> Respectfully, we disagree with the claim that the contribution of our paper is merely interpretive. While most prior work in the fairness evaluation literature focuses on quantifying disparities between **two** groups and on **worst-case** scenarios, our framework generalizes this perspective to accommodate multiple sensitive groups and to account for the **entire** value distribution. Methodologically, our primary contribution is to repurpose sparsity-based techniques for fairness evaluation and to establish a theoretical connection between sparsity and the existing maximum pairwise distance (MPD) metric which can be considered as one kind of sparsity metric focusing on worst-case scenario.
>
> Specifically, below is a summary of the comparisons between the MPD metric and the sparsity-based metric that we examine in the paper:
>  - **Common property.** We theoretically show that the sparsity-based metric and the MPD-based metric attain their respective maxima and minima under the same conditions (Remark 3.1 and Remark 3.2). This property explains why most bias-mitigation algorithms in our benchmark exhibit aligned trends across both metrics, as they aim to minimize disparities among group-level outputs.
>  - **Divergent behaviors.** Existing benchmarks rarely reveal divergent outcomes between the sparsity-based and MPD-based metrics, largely because they rely on a limited number of fairness groups. To overcome this limitation, we first provide a theoretical analysis showing that the sparsity-based metric is sensitive to variations in intermediate values, whereas MPD captures only the maximum gap within a vector (Theorem 3.6). We then conduct intersectional fairness experiments to demonstrate the divergence that emerges when the number of fairness groups is large. Furthermore, in the revised paper, we further introduce a practical use case in Recommendation Systems, where fairness across individual users as a group becomes a natural and relevant setting for observing such divergent behaviors. Specifically, the results show that the MPD-based metric is not only highly sensitive to outliers introduced by new samples but can also become dominated by them, whereas the sparsity-based metric remains comparatively stable. Such stability is theoretically expected, given that new samples are drawn uniformly and the pretrained model remains unchanged. We update Section 5.3 and add a Discussion Section in the revised PDF with the updated content highlighted in blue.
>
> > Response to Weakness 2: Results are mostly known property...
>
> We argue that our study offers substantive new insights and theoretical contributions that extend well beyond the findings of Diao et al. (2023) [1]. While the original work introduces the PQ Index primarily as a compressibility measure for network pruning, we are the first to demonstrate its potential relevance for group fairness evaluation. As noted in line 142, Theorems 3.1 through 3.4 formally establish the conditions under which the PQ Index can be interpreted as a fairness metric, and Theorems 3.5 and 3.6 further characterize its theoretical relationships with MPD and the Gini Index. With the exception of Theorem 3.1, all remaining results, including Theorems 3.2 through 3.6 and their accompanying analyses, are original and novel contributions of this work.
>
> > Response to Weakness 3: There should be a constraint...
>
> Thanks for pointing that out. We fixed this typo in the revision to make it align with the Remark 1.
>
> > Response to Question 1: Is there a principled way to select the sparsity parameters...
>
> The general recommendation is to set $p = 1$ and $q = 2$, based on the theoretical properties established in Section 3.1. Empirically, Figure 8 (Appendix) further shows that as long as the condition $0 < p \leq 1 < q$ is satisfied, varying $p$ and $q$ influences only the scale of the metrics, while the overall alignment patterns remain unchanged.
>
> ---
> [1] “Pruning deep neural networks from a sparsity perspective”

---

### Author Response · Authors · 2025-11-29
**General Response**

We sincerely thank all reviewers for their detailed assessments and invaluable feedback. These constructive comments have been instrumental in strengthening the overall quality of our work. We hereby begin by summarizing the major contribution of our work and address common and most critical concerns raised by reviewers.

Our work presents a unified perspective on group fairness evaluation. Existing fairness metrics are often tailored to specific machine learning settings, such as regression or classification, and typically quantify disparities between two groups while emphasizing worst-case scenarios through the Maximum Pairwise Distance (MPD). Our framework generalizes this perspective by accommodating multiple sensitive groups and capturing the entire value distribution through sparsity. We are the first to establish both the ethical and theoretical connections between sparsity and existing fairness evaluation paradigms.

Through extensive experiments across diverse benchmarks, we validate the proposed theorems and analyze the conditions where the two measures yield comparable or divergent results. This work provides a novel perspective on algorithmic fairness by framing it through the lens of sparsity and social equity, offering the potential for broad impact on fairness research and practical applications.

Below are some critical concerns raised by reviewers:

- Benefits of using sparsity over MPD (reviewer Nm1h, GwDM)

Given the inherently subjective nature of fairness, we clarify that our paper does not claim the sparsity-based metric to be universally superior to existing fairness metrics. We propose it as an alternative for settings that prioritize fairness across all groups rather than worst-case outcomes. Not only are the two measures theoretically and methodologically distinct, but we also design experiments that illustrate how their behaviors diverge as the number of fairness group increases. The results in Section 5.3 show that 1) when the maximum disparity is fixed, 2) when outlier groups exist, or 3) in online evaluation settings, the two metrics can yield different outcomes. Moreover, this distinction becomes more pronounced as the number of group grows. Under such scenarios, if a user values stability and robustness in fairness assessment, the sparsity-based metric may be more suitable. Conversely, if worst-case disparity is the primary concern, MPD remains the more appropriate choice.

- Methodological novelty of the proposed framework (reviewer NYpV)

Methodologically, our primary contribution is to repurpose sparsity-based techniques for fairness evaluation and to establish a theoretical connection between sparsity and the existing MPD metric, which can be interpreted as a specific form of sparsity measure that emphasizes worst-case disparities.

Regarding their commonality, we theoretically show that the sparsity-based metric and the MPD-based metric attain their respective maxima and minima under the same conditions (Remark 3.1 and Remark 3.2). In terms of their differences, we provide a theoretical analysis demonstrating that the sparsity-based metric is sensitive to variations in intermediate values, whereas MPD captures only the maximum gap within a vector (Theorem 3.6).

- How to choose $p$ and $q$ (reviewer NYpV, 7wmZ)

The general recommendation is to set $p = 1$ and $q = 2$, based on the theoretical properties established in Section 3.1. Empirically, Figure 8 (Appendix) further shows that as long as the condition $0 < p \leq 1 < q$ is satisfied, varying $p$ and $q$ influences only the scale of the metrics, while the overall alignment patterns remain unchanged.


Based on the valuable feedback from reviewers, we have revised the manuscript to better demonstrate the divergence between MPD and sparsity-based metric. Specifically, we add:
- **Additional Experiment in Section 5.3** We further introduce a practical use case in Recommendation Systems, where fairness across individual users as a group becomes a natural and relevant setting for observing such divergent behaviors. The results show that the MPD-based metric is not only highly sensitive to outliers introduced by new samples but can also become dominated by them, whereas the sparsity-based metric remains comparatively stable. Such stability is theoretically expected, given that new samples are drawn uniformly and the pretrained model remains unchanged.
- **A dedicated Discussion Section** In this section we summarize the conditions we examine when the two metrics differ and conclude the property sparsity-based metric can offer over MPD-based metric.
- **Additional Experiment in Appendix D.3** We conduct a constrained loss optimization with different fairness metric as the regularizer. The result shows PQ Index loss can be more effectively minimized comparing to MPD and Gini Index. These findings provide preliminary evidence that may inform the development of more efficient bias mitigation algorithms in future work.

---

### Meta-Review · Area_Chair_vpNw · 2025-12-20

**Summary:**

This manuscript proposes a unified framework for evaluating group fairness by leveraging sparsity measures. The work aims to connect different existing fairness criteria through a sparsity perspective and to demonstrate that this unified view offers broad applicability across domains and fairness settings. The authors provide theoretical formulations and support their claims with a set of empirical evaluations on multiple datasets.


I argue that this manuscript presents an interesting study. However, it must be acknowledged that the current version does not yet meet the standards for acceptance. The reviewer opinions showed moderate agreement on the relevance and promise of the work, but differed in their assessment of completeness and empirical strength.  Based on the reviewers' comments, the following aspects require further improvement in future revisions, i.e.,

1) Some reviewers felt that the assumptions used in deriving the sparsity-based metrics require clearer justification and could be overly restrictive in certain real-world scenarios.

2) Multiple reviewers noted that the manuscript could benefit from additional clarity in key definitions, more intuitive explanations, and illustrative figures to communicate the core concepts.

3) The writing quality was considered uneven in places, with certain technical sections needing reorganization for better logical flow.

4) Lack of comprehensive comparison against state-of-the-art fairness metrics and mitigation strategies.

**Reviewer Concerns:**

This manuscript receives four diverse comments with three negative evaluations and one positive evaluation.  The reviewers raise several concerns about 1) the motivation of using sparsity over MPD, 2) methodological novelty of the proposed method, 3) insufficient experimental comparison and discussion. Based on the interactions, and my evaluation, I think the following concerns remain unresolved,

1. The assertions regarding the potential of sparsity-based measure regarding intersectional fairness seem vague and not clearly substantiated by the experiment (Reviewer Nm1h).

2. The advantages of Gini index needs to be in line with the research motivation of this work, and thus requires thorough discussion and analysis (Reviewer GwDM).

Overall, even though this manuscript presents an interesting study,  the current version does not yet satisfy the standards for acceptance based the evaluation from both reviewers and myself.

**Reviewer Scores:**

This manuscript receives four diverse comments with three negative evaluations and one positive evaluation. The discussion phase did not involve substantial back-and-forth between the reviewers and the authors.

---

### Decision · Program_Chairs · 2026-01-26

Reject